# DivScene: Benchmarking LVLMs for Object Navigation with Diverse Scenes and Objects

## Abstract

Object navigation in unknown environments is crucial for deploying embodied agents in real-world applications. While we have witnessed huge progress due to large-scale scene datasets, faster simulators, and stronger models, previous studies mainly focus on limited scene types and target objects. In this paper, we study a new task of navigating to diverse target objects in a large number of scene types. To benchmark the problem, we present a large-scale scene dataset, DivScene, which contains 4,614 scenes across 81 different types. With the dataset, we build an end-to-end embodied agent, NatVLM, by fine-tuning a Large Vision Language Model (LVLM) through imitation learning. The LVLM is trained to take previous observations from the environment and generate the next actions. We also introduce CoT explanation traces of the action prediction for better performance when tuning LVLMs. Our extensive experiments find that we can build a performant LVLM-based agent through imitation learning on the shortest paths constructed by a BFS planner without any human supervision. Our agent achieves a success rate that surpasses GPT-4o by over 20%. Meanwhile, we carry out various analyses showing the generalization ability of our agent.

## 1 Introduction

Object navigation has long been a challenging embodied task, where an embodied agent is required to navigate to target objects in unseen environments (Batra et al., 2020; Anderson et al., 2018). This task is fundamental to other navigation-based embodied tasks, as navigating to a target object is the agent's preliminary step in interacting with objects in a scene. As the last few years have witnessed huge progress with large-scale scene datasets (Chang et al., 2017) and faster simulators (Kolve et al., 2017), numerous methods have been proposed for better navigation, including reinforcement learning (RL) (Ye et al., 2021), imitation learning (IL) (Ramrakhya et al., 2022), semantic maps (Zheng et al., 2022), and others (De Vries et al., 2018; Chaplot et al., 2020a).

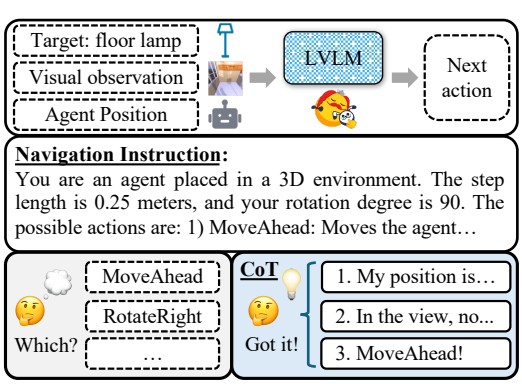

Figure 1: Illustration of our NatVLM agent. We build CoT explanation traces to help the LVLM to better grasp the rationale of object navigation.

Albeit those methods achieve state-of-the-art performance on existing object navigation tasks, they only focus on a limited set of target objects and scenes with little variety. For example, the photorealistic dataset Matterport-3D (Chang et al., 2017) only considers 21 target object types in 90 private homes. Similarly, the simulated environment ProcTHOR (Deitke et al., 2022) only considers 16 object types in four types of residential rooms (i.e., bedroom, living room, kitchen, and bathroom). With this limited scope, models often perform poorly (Zhou et al., 2023) when facing diverse unseen objects and scenes in real-world applications due to the distribution shifting.

In this paper, we study a new task of building navigational agents capable of generalizing to a wide range of unseen objects in diverse scenes. We introduce a new dataset, DivScene, that features the

most comprehensive range of scene types to the best of our knowledge. Specifically, we collect 81 scene types based on MIT Scenes Dataset (Quattoni & Torralba, 2009), which are further categorized into five big groups. Then, we use LLMs to automatically compose diverse house descriptions by adding attributes to scene types, such as "a *bakery* with tile patterned walls." We input these descriptions into the language-guided framework, Holodeck (Yang et al., 2024), to build houses automatically with the strong ability of GPT-4 (OpenAI, 2023). Finally, we compile 4,614 houses across 81 distinct scene types on the AI2THOR platform (Kolve et al., 2017). Since we take advantage of the object assets Objaverse (Deitke et al., 2023), our houses contain over 22K different kinds of objects, making them ideal for testing navigation to diverse objects.

With DIVSCENE, we proceed to build an embodied agent that can navigate to diverse objects in diverse types of houses. As aforementioned, existing works, such as RL-based (Maksymets et al., 2021) and IL-based (Ramrakhya et al., 2022) methods, only focus on learning in-distribution knowledge for limited target objects and scene types. Meanwhile, recent studies (Yu et al., 2023; Zhou et al., 2023; Dorbala et al., 2023) utilize LLMs as the planning backbone to navigate to diverse objects in a zero-shot manner, leveraging LLM's commonsense knowledge for high-level guidance. However, they still face a substantial domain gap between navigation tasks and the LLM training corpus (Lin et al., 2024). Moreover, those methods (Chen et al., 2023; Zhu et al., 2024) require a captioning model to generate textual descriptions of each scene, serving as the perception of the backbone LLMs. The captioning model might miss crucial information about objects and lead to suboptimal results, like spatial or color details.

To tackle the challenge, we propose an end-to-end object navigation agent called NATVLM (**Na**vigational Chain-of-**T**hought **VLM**), based on the large vision language model Idefics 2 (Laurençon et al., 2024). As shown in Figure 1, the model is tuned to generate the next action given the current observation, such as the the egocentric view and the agent's status. Here, we train the LVLM with imitation learning (Brohan et al., 2022) using shortest paths from a heuristic planner. To improve the accuracy, we also manually collect complex CoT explanation traces of each prediction (Mitra et al., 2023; Ho et al., 2023) to help the LVLM understand the underlying rationale behind object navigation. By tuning the LVLM, we eliminate the domain gap between navigation tasks and the pre-training corpus. Meanwhile, our navigation is performed in an end-to-end manner with perception, circumventing the captioning model.

Existing IL-based works trained their navigational agents on large corpora of human demonstrations (Brohan et al., 2022; Wei et al., 2023), which is incredibly expensive. In contrast, our study finds that imitating the shortest paths constructed by a heuristic planner can be an effective approach to training agents based on LVLMs. Specifically, the AI2THOR platform discretizes the environment into a grid map with a fixed step size. Then, we search for the shortest path from the agent to the target objects. In total, we collect about 23K shortest-path episodes in the houses from DIVSCENE, forming the DIVTRAJ dataset. The dataset contains 5,707 different kinds of target objects, significantly more than other navigation datasets.

With shortest-path episodes, we perform extensive experiments and analyses to evaluate our NATVLM agent. First, we introduce several baselines using various LLMs and VLMs (Liu et al., 2024b; Touvron et al., 2023) to establish baseline performance levels. The evaluation results show that our proposed agent can navigate to diverse objects more effectively than baselines by a large margin. We carry out thorough ablation studies to show the efficacy of CoT explanation traces in action prediction. Moreover, few-shot experiments demonstrate the robustness of our agent. Last but not least, we validate the generalization ability of our agent on two out-of-distribution datasets: ProcTHOR (Deitke et al., 2022) and iTHOR (Weihs et al., 2021).

## 2 RELATED WORK

Recent breakthroughs in large vision language models (LVLMs) have shown success in several domains (Li et al., 2023b; Wang et al., 2023; Team, 2023; Jiang et al., 2024). Our work first builds a navigational agent based on LVLMs with the following related areas of study:

**Visual Navigation:** Visual navigation is a critical task for building intelligent agents imitating human behavior. Conventional methods depend on an environment map but struggle with the low generalization ability in unseen houses (Yamauchi, 1997). Recent works (Anderson et al., 2018)

have studied more forms of the visual navigation task, like PointNav (Wijmans et al., 2019), ImageNav (Zhu et al., 2017), ObjectNav (Chaplot et al., 2020b), RoomNav (Wu et al., 2018), and visual-language navigation (Huang et al., 2023). In this work, we study the ObjectNav task and build the first end-to-end navigational agent based on LVLMs.

**Policy Learning of Object Navigation:** Zhu et al. (2017) first proposed a **reinforcement learning** (RL) method that used ResNet (He et al., 2016) to encode images. Following them, the RL-based approaches have gained attention for tackling object navigation (Anderson et al., 2018; Batra et al., 2020; Savva et al., 2017; Wahid et al., 2021; Yang et al., 2018; Druon et al., 2020; Ehsani et al., 2021). However, existing works (Ehsani et al., 2023; Guo et al., 2018) show that RL requires extensive reward shaping and is often too slow and ineffective. **Imitation learning** (Pomerleau, 1988; Zhang et al., 2018) presents a compelling alternative to RL as it reframes the task as a supervised learning problem. RT-1 and RT-2 (Brohan et al., 2022; 2023) scale up the multi-task data, focusing on object manipulation. Wei et al. (2023) also propose a prompt-guided imitation learning method. However, these methods face the tremendous cost of human demonstration. While SPOC (Ehsani et al., 2023) attempts to resolve the issue, they trained a transformer-based agent from scratch, requiring a substantial amount of training data. Meanwhile, their study is still limited to four room types in ProcTHOR (Deitke et al., 2022). Compared with those works, we fine-tune pre-trained LVLM to navigate diverse scenes and construct shortest-path trajectories without expensive cost.

**LLMs and VLMs for Visual Navigation:** With the recent advancement, LLMs and VLMs (Achiam et al., 2023; Touvron et al., 2023; Lu et al., 2024) have emerged as the foundation for solving embodied tasks (Pan et al., 2023; Majumdar et al., 2024). Researchers also explored their usage for object navigation. A few methods have employed contrastive VLMs (Radford et al., 2021; Li et al., 2022), as the visual encoders (Khandelwal et al., 2022; Majumdar et al., 2022) or object-grounding tools (Gadre et al., 2023; Dorbala et al., 2023). Yu et al. (2023) utilize LLMs as the planning backbone for object navigation in a zero-shot manner. LLMs are used as high-level planners with a captioning model to provide perception. Following them, many improvements have been proposed (Cai et al., 2024; Chen et al., 2023; Zhou et al., 2023; Shah et al., 2023). However, a substantial domain gap exists between navigation tasks and the LLM pre-training corpus (Lin et al., 2024). Meanwhile, using LLMs for navigation might lose important visual details. In contrast, we tune our end-to-end agent from an LVLM through imitation learning without those drawbacks.

**Embodied Environment:** Embodied environments are the foundation of embodied research. Existing environments are typically crafted through the manual labor of 3D artists (Deitke et al., 2020; Gan et al., 2020; Li et al., 2023a; Khanna et al., 2024; Tang et al., 2023; Xia et al., 2018), hard to scale up. Many works construct scenes from 3D scans (Savva et al., 2019; Ramakrishnan et al., 2021; Szot et al., 2021), but the scenes are not interactive. Meanwhile, ProcTHOR (Deitke et al., 2022) proposes a procedural method but only focuses on four scene types. Recently, Yang et al. (2024) introduced a language-guided system using GPT-4 to automatically generate customized scenes from textual house descriptions, called Holodeck. In this work, we collect diverse house descriptions with GPT-4, which are further used in Holodeck to collect houses.

## 3  TASK DEFINITION

In this paper, we study the object navigation task involving an agent in an environment to find the target object belonging to a given category. The object category set is denoted by $C = \{c_0, c_1, \ldots, c_m\}$, where $m$ is the number of categories. The scenes can be described by $S = \{s_0, s_1, \ldots, s_n\}$, and $n$ is the total number of scenes. In each episode, the embodied agent is initialized at a random position $p_i$ with rotation $r_i$ in a scene $s_i$. Then, the embodied agent is required to perform instance-level object navigation, where the agent receives a target object within the category $c_i$ at the position $o_i$. Thus, an episode can be represented as $E_i = \{s_i, p_i, r_i, c_i, o_i\}$. At each time step $t$, the embodied agent observes the environment and predicts the next action $a_t$. Following previous work (Yu et al., 2023; Zhu et al., 2024), the observation comprises an RGB image of the egocentric view and the agent status (i.e., its position and rotation).

Our new houses are all collected on the AI2THOR platform (Kolve et al., 2017), and thus the action space, signified as $A$, covers four actions for navigation: MOVEAHEAD, ROTATERIGHT, ROTATERIGHT, and DONE. We adopt the default settings of the AI2THOR platform. The MOVEAHEAD action moves the agent 25 centimeters, and ROTATELEFT and ROTATERIGHT actions rotate the

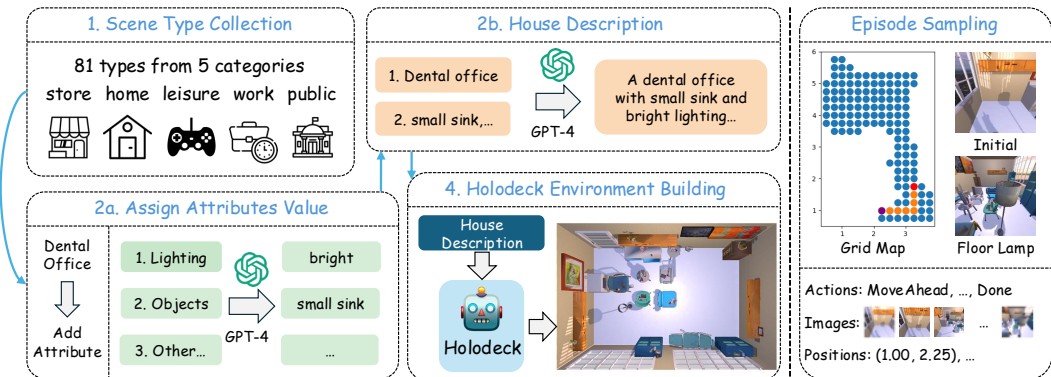

Figure 2: Data collection process. On the left, we show the process of collecting the scene dataset. We prompt GPT-4 to collect textual house descriptions and use Holodeck to build houses automatically. On the right, we show an episode built in the house. We use BFS to find the shortest path from the initial position to the target object. Then, actions and observations are collected.

agent 90 degrees. DONE is used when the agent believes it has finished the navigation task. When the agent takes the DONE action or reaches the max action limit, the episode is considered successful if the distance to the target object is less than 1.5 meters. According to this setting, an environment can be formulated as a $0.25 \times 0.25$-meter grid map of all reachable positions.

## 4 DATA COLLECTION FOR DIVSCENE AND DIVTRAJ

Existing object navigation studies only focus on limited types of scenes and objects. To fill this gap, we first curate a large-scale scene dataset DIVSCENE, featuring 81 scene types. We show the details in Figure 2. Then, 23K episodes with diverse target objects are sampled using a BFS-based planner, forming the DIVTRAJ dataset.

### 4.1 SCENE COLLECTION FOR DIVSCENE

We adopt the Holodeck (Yang et al., 2024) framework to build scenes automatically, easing human labor. Holodeck takes textual house descriptions as input and uses GPT-4 to decide the layout, styles, and object selections. To collect diverse houses, we first manually compile 81 scene types across five big groups by supplementing the MIT scene dataset (Quattoni & Torralba, 2009), like music studio and home office. We present a few houses in Figure 3 and more examples in Appendix E.

Then, we build textual house descriptions based on randomly chosen scene types by adding house attributes. We consider 12 house attributes, such as room style, users of the room, etc. We randomly sample 1-3 attributes and prompt GPT-4 to assign specific values to them. A house description is then written by GPT-4, given the scene type and attribute values. Here, we prompt GPT-4 under the in-context learning setting with the standard instruction-then-exemplar prompt (West et al., 2021):

<INSTRUCTION>
<EX$_1$-IN-TYPE><EX$_1$-IN-ATTR> <EX$_1$-OUT>
. . .
<EX$_N$-IN-TYPE><EX$_N$-IN-ATTR> <EX$_N$-OUT>
<EX$_{N+1}$-IN-TYPE><EX$_{N+1}$-IN-ATTR>

where <INSTRUCTION> describes the task of writing house descriptions. <EX$_i$-IN-TYPE> and <EX$_i$-IN-ATTR> are the selected scene type and attributes, with output <EX$_i$-OUT> being an exemplar description. Concrete prompts are shown in Appendix A.2, and we provide GPT-4 with $N = 5$ exemplars to generate the description for the final test example. Also, we leave the full details of scene types and attributes in Appendix A.1

To encourage diversity, a new description is included only when its ROUGE-L similarity (Lin, 2004) with any existing description is less than 0.8. Invalid generations are also identified and filtered out based on heuristics (e.g., invalid output format). We show more details in Appendix A.3. After we

collect abundant scene descriptions, we input each of them into the Holodeck framework and build a dataset of 4,614 scenes named DIVSCENE.

## 4.2 EPISODE COLLECTION FOR DIVTRAJ

We build a BFS-based planner and design a pipeline to produce expert trajectories of shortest paths. Our method has three steps to sample each episode. First, we sample an initial position and target object. Then, we use our planner to find the shortest path between them. Finally, we obtain the action sequence and corresponding environment observations. We show an example in Figure 2.

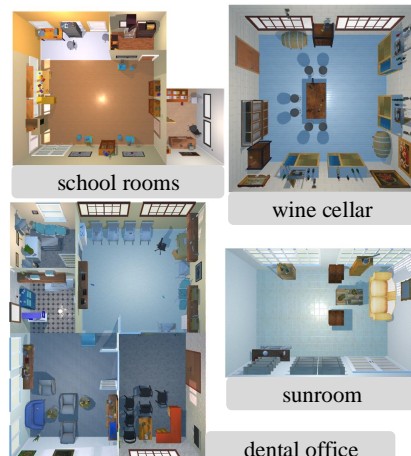

Figure 3: Examples of houses in DI-VSCENE with different scene types.

As discussed in Section 3, houses in AI2THOR are discretized as grid maps of the fixed step size. We randomly sample an initial position from the grip map for the agent. Then, a target object is randomly sampled from all available objects in the environment. To encourage diversity, objects of the same type as those sampled in previous episodes are removed from the pool when sampling for new episodes in the same house. We also impose that the target objects are between 0.3 m and 2.0 m in height to ensure they are observable to the agent.

Second, we build a planner capable of navigating multi-room cluttered environments with various obstacles. The planner stems from the ground truth information available in AI2-THOR, like reachable positions and objects' coordinates. We use Breadth-First Search (BFS) on the grid map to find the shortest path from the initial position to the target object. We show more details of BFS in Appendix A.4. We stress that ground truth information is not available to agents at inference time and is only used to produce expert episodes for training.

With the shortest path, we then derive the sequence of actions needed to achieve navigation. Basically, between two adjacent positions in the shortest path, we add a MOVEAHEAD action if the agent's rotation remains unaltered. Otherwise, we first use ROTATERIGHT or ROTATELEFT to adjust the orientation and then add a MOVEAHEAD action (more details in Appendix A.5). Thus, we obtain an action sequence that can steer the agent to the target object. We execute all actions in AI2THOR and collect observations at each step, including the agent's status and egocentric images.

## 4.3 STATISTICS OF BOTH DATASETS

In DIVSCENE, we collected 4,614 houses across 81 scene types. To the best of our knowledge, this dataset covers the widest range of scene types. We show the diversity of our collected houses in Figure 6 by plotting most types under each category. Thanks to Objaverse assets (Deitke et al., 2023), the collected houses contain objects from 22,696 different types, including very common objects such as fridges, beds, shelves, and sofas, and rare objects such as multicolored bookshelf and vintage wooden bench.

Then, we randomly pick one house from each scene type to build a test set of 81 houses. Similarly, we randomly select 27 houses from distinct scene types as the validation set. Thus, DIVSCENE is divided into training, validation, and test sets, covering 4506, 27, and 81 houses. On the training set, we sample five episodes in each scene with different target objects. Four episodes are selected in each house in evaluation sets to balance the evaluation efficiency and accuracy. In total, the DIVTRAJ dataset contains 22962 episodes and 5707 different kinds of target objects. More statistics of the DIVSCENE and DIVTRAJ are shown in Appendix A.6.

## 5 NATVLM

Figure 1 illustrates the datils of the proposed NATVLM model. Here, we build an end-to-end agent that imitates shortest path experts in AI2THOR. Our agent is tuned from the LVLM, Idefics 2 (Laurençon et al., 2024), in an end-to-end manner. It takes an environment observation at each

time step $t$ and generates the next action $a_t$, following the same format as instruction tuning. Here, we discuss the collection of instructions and corresponding responses.

## 5.1 INSTRUCTIONS COMPILATION

We manually build instructions for each time step in an episode, which contains four parts. First, the instruction provides a brief introduction to the object navigation task, such as possible actions and step length. Then, we add the episode-specific information, including the target object, its positions, the agent's position and rotation, and visual observation. Third, we provide the agent's positions and actions for the recent $M = 8$ steps, along with the visual observations from the recent $K = 4$ steps, to help the agent make better decisions. Finally, the instruction asks the model to predict the next action by considering the agent's position and visual observation in accordance with the CoT explanation in the responses. We show the specific instruction template in Appendix B.1.

## 5.2 RESPONSE COLLECTION WITH COT EXPLANATION TRACES

We tune the LVLM to generate the next action when instructions are given. In the model, we encode actions as natural language and use the model's generative capabilities to decode the next action. Nonetheless, we found that merely requiring LVLMs to output the next action leads to unsatisfactory results. The model only grasps the surface-level styles of the prompt but misses the underlying rationales of object navigation.

To enhance the agent's understanding of navigation rationale, we manually build responses with CoT explanation traces during the navigation process. The structure of the responses covers three steps. In the first step, we have the agent compare its current position with the target and determine whether it needs to move forward or take another action. After this, the agent is asked to check the obstacles in the visual observation to see whether it needs to rotate. In the last step, the agent gives the final decision based on the analyses in the first two steps. In contrast to writing explanations with LLMs (Mitra et al., 2023), we manually write the prompt template of explanation traces for the MOVEAHEAD and DONE actions, leaving the position information to be filled in with coordinates at each step. For the rotation actions (i.e., ROTATERIGHT and ROTATELEFT), we identify very common scenarios during the navigation and design the explanation trace template for each of them. Then, we introduce a few postprocessing steps, like action balancing and conflict filtering. We show concrete prompts and postprocessing details in Appendices B.2 and B.3, respectively.

## 6 EXPERIMENT

In this section, we conduct extensive experiments to compare our framework with various baselines.

## 6.1 DATASET AND METRICS

We conduct our experiments on DIVTRAJ with the statistics shown in Section 4.3. Our agent NATVLM and baselines navigate in a given house until the model chooses the DONE action or reaches the max action limit, 200 in our experiments. We report the metrics Success Rate (SR), Success weighted by Path Length (SPL (Anderson et al., 2018)), and Success weighted by Episode Length (SEL (Eftekhar et al., 2023)). An episode is considered successful when the target object appears in the agent egocentric observation and is less than 1.5 meters away. Specifically, SR and SPL are computed as $\frac{1}{N}\sum_{i=1}^{N} S_i$ and $\frac{1}{N}\sum_{i=1}^{N} S_i \frac{l_i}{max(l_i, p_i)}$, where $N$ is the number of episodes, $S_i$ is the indicator of success, $l_i$ is the length of the shortest path, and $p_i$ is the length of the predicted trajectory. For SEL, we replace $l_i$ and $p_i$ with the action number of the shortest and predicted paths.

## 6.2 BASELINE METHODS

We compare our embodied agent with four baselines:

**Blind LLMs:** The first baseline we test is the text-only LLM agent that simply predicts the next action based on the textual instruction without considering any visual information. This agent references how far we can get solely using prior world knowledge and random guessing. For the LLM

Table 1: Performance of NATVLM and baselines on the validation and test sets of DIVTRAJ. We also provide the performance on all evaluation data. The highest scores are bolded. In the "LLMs w/Captions" baseline, we use Llava 1.5 as the captioning model.

| Methods | Backbone | Valid | | | Test | | | All | | |
|---|---|---|---|---|---|---|---|---|---|---|
| | | SR | SPL | SEL | SR | SPL | SEL | SR | SPL | SEL |
| **Random** | - | 9.26 | 8.19 | 9.26 | 6.79 | 5.77 | 6.79 | 8.03 | 6.98 | 8.03 |
| **Blind LLMs** | Llama 2 (7B) | 8.33 | 7.26 | 3.64 | 9.57 | 7.63 | 6.28 | 8.95 | 7.45 | 4.96 |
| | Llama 2 (13B) | 9.26 | 7.69 | 3.12 | 10.19 | 8.62 | 4.14 | 9.72 | 8.15 | 3.63 |
| | Llama 3.1 (8B) | 11.11 | 9.40 | 5.56 | 12.04 | 9.50 | 6.19 | 11.57 | 9.45 | 5.88 |
| | Mistral (7B) | 8.33 | 7.16 | 4.13 | 9.88 | 7.89 | 3.78 | 9.11 | 7.53 | 3.96 |
| **LLMs w/ Captions** | Llama 2 (7B) | 11.11 | 9.30 | 5.06 | 12.96 | 10.90 | 8.19 | 12.04 | 10.10 | 6.62 |
| | Llama 2 (13B) | 9.26 | 7.56 | 4.03 | 12.35 | 9.93 | 5.95 | 10.80 | 8.74 | 4.99 |
| | Llama 3.1 (8B) | 12.96 | 10.73 | 2.75 | 16.67 | 13.50 | 6.28 | 14.82 | 12.12 | 4.52 |
| | Mistral (7B) | 11.11 | 9.65 | 3.43 | 11.76 | 9.65 | 2.72 | 11.43 | 9.65 | 3.07 |
| **Open LVLMs** | Qwen-VL (7B) | 10.19 | 8.75 | 9.14 | 7.41 | 6.05 | 6.66 | 8.80 | 7.40 | 7.90 |
| | Llava 1.5 (7B) | 12.04 | 10.07 | 9.88 | 12.35 | 10.03 | 10.30 | 12.20 | 10.05 | 10.09 |
| | Llava 1.5 (13B) | 12.04 | 10.50 | 11.05 | 10.62 | 8.73 | 9.75 | 11.33 | 9.62 | 10.40 |
| | Idefics 2 (8B) | 21.30 | 17.88 | 14.31 | 20.68 | 17.18 | 16.49 | 20.99 | 17.53 | 15.40 |
| **API LVLMs** | GPT-4v | 33.33 | 28.79 | 18.81 | 32.10 | 26.39 | 18.26 | 32.72 | 27.59 | 18.54 |
| | GPT-4o | 37.04 | 31.82 | 29.47 | 38.27 | 31.74 | 27.92 | 37.66 | 31.78 | 28.70 |
| **NATVLM (Ours)** | Idefics 2 (8B) | **57.41** | **47.84** | **47.90** | **54.94** | **44.45** | **45.83** | **56.17** | **46.15** | **46.86** |

choice, we evaluate Llama 2 (7B, 13B) (Touvron et al., 2023), Llama 3.1 (8B) (Dubey et al., 2024), and Mistral (7B) (Jiang et al., 2023).

**Socratic LLMs w/ Image Captions:** This is the simplest agent that leverages perceptual information. Here, we use an image captioning model to convert visual observations into natural language. These captions provide a language description of egocentric images, allowing LLMs to obtain the content of perceptual information. Here, we employ Llava 1.5 (Liu et al., 2024a) as the captioning model while using the same LLMs as those in the "Blind LLM" baseline.

**Open-Source LVLMs:** The most generic agent for navigation is one that can directly process textual instructions and visual observations. Thus, we directly test recent open-source LVLMs without any further tuning, which are capable of processing images in addition to textual queries. Here, the single image LVLM we test is Llava 1.5 (7B, 13B) (Liu et al., 2024b;a). Meanwhile, we test Qwen-VL (7B) (Bai et al., 2023) and Idefics 2 (8B) (Laurençon et al., 2024) for handling multiple images.

**API-based LVLMs:** In addition to open-source LVLMs, we also evaluate closed-source ones, including GPT-4v (OpenAI, 2023) and GPT-4o (OpenAI, 2024). They can process multiple images and achieve state-of-the-art performance in multimodal tasks.

## 6.3 MAIN EVALUATION

We present the results of our model NATVLM and baselines on the validation and test sets in Table 1. We also include the performance of selecting a random action at each step (i.e., **Random**) as a reference. In general, our embodied agent NATVLM achieves the best performance on object navigation, exceeding the performance of all baselines by a large margin. For example, our model can successfully navigate to 57.41% of episodes on the test set, increasing by about 20% compared to the GPT-4o baseline. Meanwhile, according to the higher SPL and SEL scores on both test and validation sets, our model can navigate to target objects with better efficiency.

For the baselines, the blind LLMs achieve performance slightly higher than the random results. For example, LLama 3.1 (8B) achieves a success rate of 11.57%, about 4 points higher than the random guess. In the meantime, we observe that the performance of all LLMs only improved marginally when we added the captioning model to provide additional perceptual information. This result shows that the captioning model can miss important image content details, leading to unsatisfactory improvement. On the other hand, we find that LVLMs can achieve the best results across all baselines.

Table 2: The ablation study of our agent NATVLM. $\Delta_*$ columns indicate the score difference of each metric on the test set. We remove the explanation trace and test different methods of position comparisons. We bold the highest scores except for the gold label test ($\diamond$ w Gold Label).

| Methods | Valid | | | Test | | | Test Diff | | |
|---|---|---|---|---|---|---|---|---|---|
| | SR | SPL | SEL | SR | SPL | SEL | $\Delta_{SR}$ | $\Delta_{SPL}$ | $\Delta_{SEL}$ |
| Ours | **57.41** | **47.84** | **47.90** | **54.94** | **44.45** | 45.83 | - | - | - |
| $\diamond$ w/o ET | 29.63 | 25.01 | 23.18 | 26.54 | 22.11 | 21.42 | ↓28.40 | ↓22.34 | ↓24.41 |
| $\diamond$ w/o ET & w Gold | 28.70 | 24.12 | 23.38 | 30.86 | 25.46 | 25.58 | ↓24.08 | ↓18.99 | ↓20.25 |
| $\diamond$ w Gold Label | 59.26 | 49.01 | 51.33 | 62.96 | 50.54 | 54.12 | ↑8.02 | ↑6.09 | ↑8.29 |
| $\diamond$ w Diff-EQ | 54.63 | 45.02 | 46.48 | 54.32 | 43.59 | **46.84** | ↓0.62 | ↓0.86 | ↑1.01 |

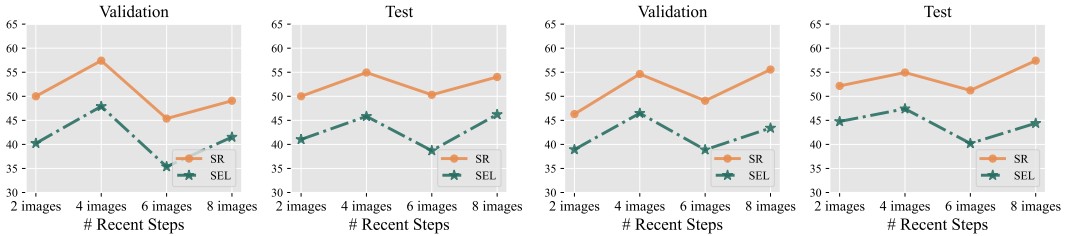

(a) Performance of the default prompt.  (b) Performance of the difference-equation prompt

Figure 4: Design investigation. We provide different numbers of recent visual observations with the embodied agent. Besides the default prompt we use, we also evaluate the difference-equation prompt. See scores of all metrics in Appendix C.2

For example, the closed-source VLMs, GPT-4v and GPT-4o, can successfully navigate to target objects in more than 30% cases. In addition, Idefics 2 (8B) attains success rates exceeding 20% on both validation and test sets.

## 6.4 ABLATION STUDY

To better understand the role of CoT explanation traces in our agent NATVLM, we conduct two ablation studies to analyze its contribution. First, we verify the efficacy of CoT explanation traces. Then, we analyze different ways to compare the positions of the agent and target object. The results of experiments are shown in Table 2.

First, we remove the whole explanation traces in the response of the instruction tuning data ($\diamond$ w/o ET). Thus, we fine-tune the agent to only generate the next action. From the result, we can find that the performance of our agent drastically drops. This verifies that explanation traces can help the LVLM to better understand the underlying rationales of object navigation. Subsequently, we further enhance the agent's input by providing the gold label indicating the positional difference between the agent and the target object ($\diamond$ w/o ET & w Gold). As shown in Table 2, we can observe that the gold labels cannot help much as the performance only fluctuates somewhat.

Then, we study the effects of the position comparison, a crucial component and the initial step in our explanation traces. The default prompt is to directly generate the position difference: "the difference to the target object is **[position_diff]**" as shown in Table 10, where **[position_diff]** is a placeholder. First, we provide the global label of positional differences in the input instructions ($\diamond$ w Gold Label). The results in Table 2 show that providing global labels can improve the performance of our agent. This observation suggests that our agent may occasionally compute the positional difference with inaccuracies, as providing the gold label can lead to a performance increase. Next, we test the difference-equation prompt ($\diamond$ w Diff-EQ), where we fine-tune our agent to generate an equation for computing the positional difference. However, writing the equation of computation only leads to small variations in performance. All the abovementioned prompts are shown in Appendix C.1.

Table 3: Hyperparameter investigation. Our agents receive various numbers of recent actions and positions within the default prompt. See results of difference-equation prompt in Appendix C.3.

| Number | Valid | | | Test | | | All | | |
|---|---|---|---|---|---|---|---|---|---|
| | SR | SPL | SEL | SR | SPL | SEL | SR | SPL | SEL |
| 4 steps | 46.30 | 38.66 | 39.82 | 45.99 | 37.45 | 39.60 | 46.14 | 38.05 | 39.71 |
| 8 steps | **57.41** | **47.84** | **47.90** | **54.94** | **44.45** | **45.83** | **56.17** | **46.15** | **46.86** |
| 12 steps | 42.59 | 35.92 | 36.43 | 50.93 | 41.15 | 43.20 | 46.76 | 38.53 | 39.81 |
| 16 steps | 51.85 | 43.01 | 42.72 | 53.40 | 43.03 | 44.08 | 52.62 | 43.02 | 43.40 |

Table 4: Few-shot learning ability. We test our agent with different proportions of training data. Besides the default prompt, we also evaluate the difference-equation prompt in Appendix C.4.

| % Data | Valid | | | Test | | | Test Diff w GPT-4o | | |
|---|---|---|---|---|---|---|---|---|---|
| | SR | SPL | SEL | SR | SPL | SEL | $\Delta_{SR}$ | $\Delta_{SPL}$ | $\Delta_{SEL}$ |
| 20 | 37.04 | 30.86 | 30.22 | 38.89 | 31.40 | 32.13 | ↑0.62 | ↓0.34 | ↑4.21 |
| 40 | 33.33 | 27.95 | 27.13 | 38.27 | 31.00 | 30.79 | ↓0.00 | ↓0.74 | ↑2.87 |
| 60 | 49.07 | 40.68 | 42.01 | 52.12 | 42.25 | 44.40 | ↑13.85 | ↑10.51 | ↑16.48 |
| 80 | 50.00 | 41.46 | 43.36 | 49.69 | 40.26 | 42.49 | ↑11.42 | ↑8.52 | ↑14.57 |
| 100 | **57.41** | **47.84** | **47.90** | **54.94** | **44.45** | **45.83** | ↑16.67 | ↑12.71 | ↑17.91 |

## 6.5 DESIGN INVESTIGATION

In this experiment, we thoroughly investigate a few design decisions regarding the image number and step number of recent positions and actions. Besides the default prompt structure, we also conduct experiments with the difference-equation prompt, which is introduced as "◇ w Diff-EQ" in the ablation study.

The default design provides NATVLM with four images. In addition, we test the agent's performance using different numbers of input images, including 2, 6, and 8 images. The results are plotted in Figure 4. First, we observe a decline in the agent's performance when provided with only two images, showing that using fewer images leads to worse performance. For instance, our agent finishes the navigation successfully only 50% of the time, underperforming the 4-image baseline. Moreover, increasing the number of images to 6 or 8 does not result in further performance variations. Thus, we choose to provide our agent with four images for the tradeoff between accuracy and efficiency.

We also investigate the effect of recent positions and actions on navigation performance. By default, we provide the agent with information about the recent 8 steps. Then, we test the performance when we provided the positions and actions of 4, 12, and 16 steps. As illustrated in Table 3, a substantial performance improvement is evident when increasing the number of steps from 4 to 8. However, further increases in step count yield limited returns. Thus, we provide our agent with 8 steps.

## 6.6 FEW-SHOT LEARNING ABILITY

Our agent undergoes instruction tuning with an extensive training dataset, DIVTRAJ. To assess its generalization capabilities, we design a few-shot experiment to confirm its ability to generalize with fewer data. While the original training data contains five episodes per house, we train the model with only 1, 2, 3, and 4 episodes, representing 20%, 40%, 60%, and 80% of the full data. The results are shown in Table 4. For clarity, we also compare our models with the GPT-4o baseline. We can find that our agent has strong few-shot learning abilities. With only 20% percent training data, our agent can perform similarly to GPT-4o and generalize well on unseen houses. Meanwhile, the results demonstrate a gradual performance improvement as we incrementally increase the data volume, with performance gains plateauing at approximately 80% of the full dataset.

## 6.7 ZERO-SHOT TRANSFER LEARNING

To further verify the generalization ability of NATVLM, we test it on other house datasets with limited scene types: iTHOR (Weihs et al., 2021) and ProcTHOR (Deitke et al., 2022). Both datasets

Table 5: Performance of zero-shot transfer learning on iTHOR and ProcTHOR. NATVLM outperforms all baselines by a large margin.

| Models | iTHOR | | | ProcTHOR | | |
|---|---|---|---|---|---|---|
| | SR | SPL | SEL | SR | SPL | SEL |
| Qwen-VL (7B) | 23.67 | 19.96 | 19.27 | 10.83 | 9.04 | 6.28 |
| Llava 1.5 (7B) | 24.12 | 20.32 | 20.34 | 16.04 | 13.53 | 14.02 |
| Llava 1.5 (13B) | 19.47 | 16.21 | 17.48 | 13.12 | 11.07 | 11.85 |
| Idefics 2 (8B) | 28.54 | 23.39 | 18.49 | 17.29 | 14.33 | 11.05 |
| NATVLM ◇ w/o ET | 38.27 | 32.15 | 31.43 | 31.25 | 26.87 | 26.20 |
| NATVLM | **72.79** | **59.34** | 59.28 | 53.12 | 44.37 | 43.04 |
| NATVLM ◇ w Diff-EQ | 72.32 | 58.98 | **62.35** | 54.59 | 45.53 | 46.80 |

encompass four distinct scene types: bedrooms, living rooms, kitchens, and bathrooms. There are 30 rooms in iTHOR for each scene type, which is designed by professional 3D artists. Meantime, ProcTHOR is a procedural house-generation system that constructs 10,000 unique houses automatically. Similar to iTHOR, we sample 30 houses for each scene type from ProcTHOR. Four episodes are sampled in each house to evaluate our agent.

We directly use NATVLM tuned on the DIVTRAJ dataset to test this zero-shot transferring ability. Since we do not tune hyper-parameters on these datasets, we treat each dataset as a test set without any validation set. The results are shown in Table 5. We also report the performance of open-source VLMs and some ablated models as baselines. The results show that our agent surpasses all the baselines on both datasets. Such improvements indicate that our framework has a strong ability to generalize in other environments.

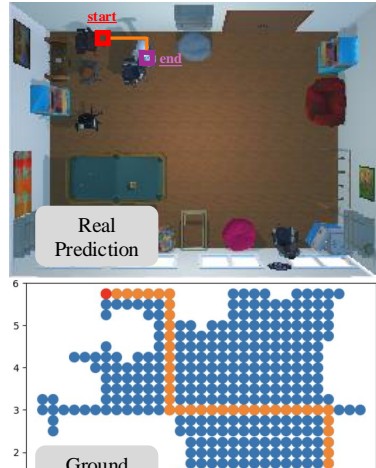

## 6.8 CASE STUDY

In Figure 5, we provide an example of error analysis of our agent. The upper image shows the predicted path in the real scene, and the lower image shows the ground truth in the corresponding grid map. Here, the agent needs to find a soda can after walking through the whole game room. There are 46 actions in the shortest path. Instead of heading towards the goal, the agent just meanders in a limited area. This shows that our agent cannot finish the navigation with a long trajectory. One possible explanation could be the constraint of only providing the agent with recent historical information.

Figure 5: Error analysis. The agent needs to find a soda can in the lower right corner of a game room.

## 7 CONCLUSION

In this paper, we study a new task of building navigational agents capable of generalizing to a wide range of unseen objects in diverse scenes. For benchmarking, we collect a large-scale scene dataset, DIVSCENE, featuring 81 different scene types. Then, we build the end-to-end agent, NATVLM, which is fine-tuned from an LVLM. The agent is trained through imitation learning on the shortest paths generated by a BFS-based planner. With the planner, we collect over 22K episodes to form the DIVTRAJ dataset. In the training data, we also build CoT explanation traces to help the agent better grasp the underlying rationale of navigation. Extensive experiments show that our agent can navigate to diverse objects in diverse scenes significantly better than baselines. We also conduct various analyses to show the generalization ability of our agent. For future work, a promising direction is to expand the memory capacity of LVLMs to enable navigation over longer horizons.

ETHICS STATEMENT

Our scene dataset DIVSCENE is built upon the publicly available AI2THOR platform (Kolve et al., 2017) and the Holodeck framework (Yang et al., 2024). Then, we further extend our experiments on iTHOR (Weihs et al., 2021) and ProcTHOR (Deitke et al., 2022), both of which are open-source datasets.

REPRODUCIBILITY

To reproduce our experiments, we add a copy of the code with readme and data examples in the supplemental materials. The datasets and models developed in this study will be released upon acceptance, including DIVSCENE, DIVTRAJ, and NATVLM. Meanwhile, we provide the full implementation details of our NATVLM agent and baselines in Appendix D, including the learning rate, batch size, hard device, API access, etc. Meanwhile, we show the whole data postprocessing details in Appendices A.3 and B.3.

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

Table 6: The categories and scene types used in our DIVSCENE dataset. In total, there are 5 categories and 81 scene types in our dataset.

| Category | Scene Type |
|---|---|
| **Store** (16 types) | bakery, grocery store, clothing store, deli, laundromat, jewellery shop, bookstore, video store, florist shop, shoe shop, toy store, furniture store, electronics store, craft store, music store, sporting goods store |
| **Home** (21 types) | bedroom, nursery, closet, pantry, children room, lobby, dining room, corridor, living room, bathroom, kitchen, wine cellar, garage, sunroom, cabinet, study room, apartment, home office, basement, attic, laundry room |
| **Public spaces** (9 types) | prison cell, library, waiting room, museum, locker room, town hall, community center, convention center, recreation center |
| **Leisure** (14 types) | buffet, fast-food restaurant, restaurant, bar, game room, casino, gym, hair salon, arcade, spa, concert hall, ski lodge, lounge, club |
| **Working place** (21 types) | hospital room, kindergarten, restaurant kitchen, art studio, classroom, laboratory, music studio, operating room, office, computer room, warehouse, greenhouse, dental office, TV studio, meeting room, school room, conference room, factory floor, call center, reception area, nursing station |

Table 7: The 12 attributes we used to collect house descriptions. We also provide an example value of each attribute.

| Attribute | Example Value | Attribute | Example Value |
|---|---|---|---|
| Room Style | victorian, rustic | Flooring | soft and cushioned, hard |
| Objects in the Room | computers, desks, chairs, servers | Theme | industrial, contemporary |
| Number of Rooms | single room | Lighting | bright, warm ambient |
| Configurations | individual cubicles | Window | small, slightly slanted |
| Users of the Room | children of various ages | Room Size | spacious, medium-sized |
| Era | contemporary, modern | Wall Treatment | artistic paintings, calming color |

## A  DIVSCENE AND DIVTRAJ DETAILS

### A.1  SCENE TYPE DETAILS AND ATTRIBUTE LIST

This section lists all scene types we collected by complementing the MIT scene dataset. In total, there are 81 scene types across five different categories, as shown in Table 6.

To collect diverse house descriptions, we add various attributes to a randomly sampled scene type. Here, we consider 12 different attributes as shown in Table 7. We ask GPT-4 to assign a value to each attribute and write a house description.

### A.2  TEXTUAL HOUSE DESCRIPTION PROMPT

We use in-context learning to prompt the GPT-4 to write textual house descriptions of 81 different scene types. Here, we show the concrete prompt we used in Table 8. We randomly sample 1-3 house attributes and ask GPT-4 to assign a value to them. Then, a house description is written based on the given scene type and attribute values. We use five exemplars in the in-context learning setting.

### A.3  POSTPROCESSING OF TEXTUAL HOUSE DESCRIPTION

After we collect textual house descriptions from GPT-4, we also introduce three filters to ensure their diversity and quality. First, we introduce a ROUGE-L filter. A new textual description is added only when its ROUGE-L similarity with any existing description is below 0.8, following previous works (Wang et al., 2022; 2024). Second, if we cannot find the values of given attributes in the first step of the output, we remove the example. Third, if we cannot find the house description in the

Table 8: The prompt we used to collect textual house descriptions using GPT-4. Here, we use 5 exemplars in the in-context learning. We show one example here for saving space.

---

**Task Instruction:** Create a detailed and fluent description for a house based on the given scene type and features in two steps. Step 1: provide the value of each feature. Step 2: write a short phrase to describe the scene type with the values.

---

**Exemplar1 Input:** The given house type is "arcade." The feature list is: "(1) Objects in the room."
**Exemplar1 Output:** Step 1: (1) a pool table\n Step 2: An arcade with a pool table

---

**Following Exemplars:** Exemplar 2, ..., Exmplar 5

---

**Testing Input:** The given house type is "office." The feature list is: "(1) Number of Rooms (2) Users of the Room (3) Configurations."

---

second step in the output from GPT-4, we remove the example. The last two steps mean that the output does not follow the output format specified in the instructions and exemplars (see Table 8).

A.4    BFS-BASED PLANNER

We use a BFS-based planner to find the shortest path from the initial position to a target point on the grid map. Notice that the target object is not necessarily anchored to a point on the grid map for realism. Thus, we find the grid point nearest to the target object as the destination of the navigation.

The algorithm is shown in Algorithm 1, which is based on a priority queue. We design the BFS-based planner to pick the path with the fewest rotations to make it easier for LLMs to imitate. In detail, we add more costs when rotation changes since the agent needs one more rotation action before moving ahead, as shown at the 12th line in Algorithm 1.

---

**Algorithm 1** BFS Search for Shortest Paths

---

1: **procedure** BFS_SEARCH($reachable\_pos, start\_point, end\_point, start\_rotation$)
2:     Initialize priority queue $Q$, distance map $d\_map$, and parent map $p\_map$
3:     Enqueue $(0, start\_point, start\_rotation)$ to $Q$
4:     **while** $Q$ not empty **do**
5:         $(cost, current, rotation) \leftarrow$ Dequeue from $Q$
6:         **if** $current = end\_point$ **then**
7:             **return** ReconstructPath($p\_map, current$)
8:         **end if**
9:         **for** $r$ in {North, East, South, West} **do**                    ▷ Four cardinal directions
10:            $neighbor \leftarrow$ GetNeighbor($current, r$)
11:            **if** $neighbor \in reachable\_pos$ **then**
12:                $newCost \leftarrow cost +$ (1 if $r = rotation$ else 2)   ▷ More cost if rotation changed
13:                **if** $neighbor$ not visited or $newCost < d\_map[neighbor]$ **then**
14:                    $d\_map[neighbor] = newCost$                    ▷ Update distance
15:                    $p\_map[neighbor] = current$                    ▷ Update parent
16:                    Enqueue $(newCost, neighbor, r)$ to $Q$
17:                **end if**
18:            **end if**
19:        **end for**
20:    **end while**
21:    **return** No path found
22: **end procedure**

---

A.5    ACTION DERIVATION ALGORITHM

We show the action derivation algorithm in Algorithm 2. Between two adjacent positions in the shortest path, we add a MOVEAHEAD action if the agent's rotation remains unaltered. Otherwise, we first use ROTATERIGHT or ROTATELEFT to adjust the orientation and then add a MOVEAHEAD

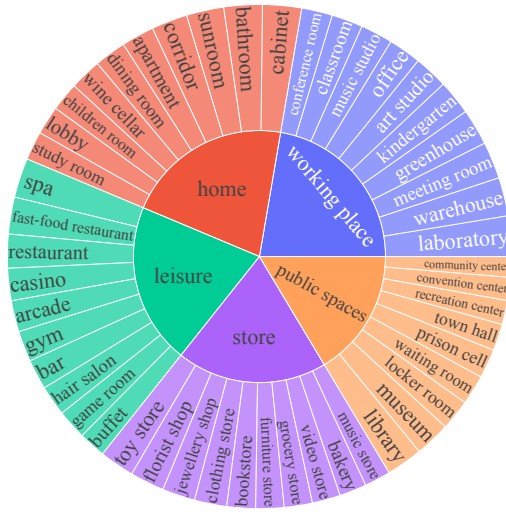 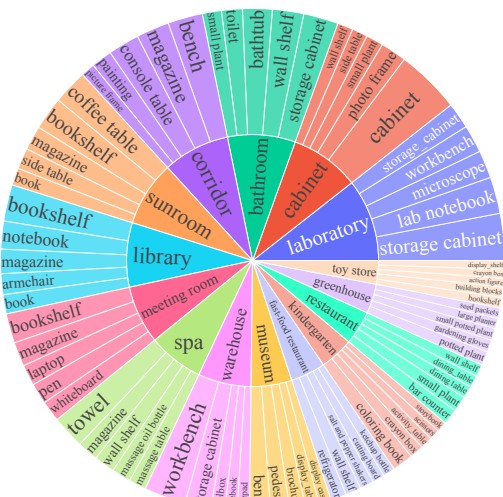

Figure 6: The top 10 most common room types (outer circle) under each room category (inner circle) in the collected houses.

Figure 7: The top 15 most common scene types (inner cycle) and their top 5 target object types (outer cycles).

action. After reaching the target object, we then rotate the agent so that the object is approximately centered in the agent's egocentric view.

## A.6 DATA DIVERSITY

To study what types of scenes are gathered under each category, we identify the category-type structure of houses in DIVSCENE. We plot the top 10 most common scene types under each category in Figure 6. Then, we study the diversity of target objects in the episodes we sampled in DIVTRAJ. We plot the top 15 most common scene types and their top 5 target object types in Figure 7. Overall, we see quite diverse scenes and target objects in our datasets.

---

**Algorithm 2** Get a Sequence of Actions from a Shortest Path

---

1: **procedure** GET_ACTION_SEQUENCE($shortest\_path, start\_rotation, target\_object$)
2:     $agent\_rotation \leftarrow start\_rotation$
3:     Initialize empty $action\_list$
4:     $prior\_p \leftarrow shortest\_path[0]$
5:     **for** each $current\_p$ in $shortest\_path[1:]$ **do**
6:         $path\_rotation \leftarrow$ compute_path_rotation($current\_p, prior\_p$)
7:         **if** $path\_rotation \neq agent\_rotation$ **then**
8:             Append appropriate rotation action(s) to $action\_list$    ▷ RotateRight or RotateLeft
9:             $agent\_rotation \leftarrow path\_rotation$
10:         **end if**
11:         Append "MoveAhead" to $action\_list$
12:     **end for**
13:     $object\_rotation \leftarrow$ compute_object_rotation($target\_object, shortest\_path[-1]$)
14:     **if** $object\_rotation \neq agent\_rotation$ **then**
15:         Append final rotation action(s) to $action\_list$    ▷ Adjust the view for the target object
16:     **end if**
17:     Append "Done" to $action\_list$
18:     **return** $action\_list$
19: **end procedure**

Table 9: Instruction Template we used to fine-tune the LVLM: Idefics 2. There are four steps in the template, and we leave step-wise and episode information with placeholders. **[obj_type]** and **[obj_pos]** are the type of target object and its location. On the grid map, we also provide the nearest point on the grid map **[grid_obj_pos]** with a rotation **[grid_obj_rotation]**. **[agent_pos]** and **[agent_rotation]** are the agent's position and rotation. There are also placeholders for the recent status: **[recent_agent_pos]**, **[recent_agent_rotation]**, **[recent_agent_image]**, and **[recent_action]**. Notice that we provide the information of the recent 8 steps and only provide the recent 4 images for inference efficiency.

---

**1. Brief Introduction:** You are an agent placed in a 3D environment. Your step length is 0.25 meters, and your rotation degree is 90.
The possible actions are:
1. MoveAhead: Moves the agent forward by 0.25 meters in the direction it is currently facing. For example, if the agent is at (x, y) facing 0 degrees (north), MoveAhead will result in (x, y + 0.25). If the agent is facing 90 degrees (east), MoveAhead will result in (x + 0.25, y). If the agent is facing 180 degrees (south), MoveAhead will result in (x, y - 0.25). If the agent is facing 270 degrees (west), MoveAhead will result in (x - 0.25, y).
2. RotateRight: Rotate right for 90 degrees (clockwise).
3. RotateLeft: Rotate left for 90 degrees. (counterclockwise).
4. Done: Indicate that you are near to the target object and finish the task.

---

**2. Episode-Specific Information:** You need to find a **[obj_type]** at the position **[obj_pos]**. To achieve this, we recommend you move to the position **[grid_obj_pos]** with a rotation of **[grid_obj_rotation]**.
Currently, you are at **[agent_pos]** with a rotation of **[agent_rotation]**.

---

**3. Status of Recent Steps:** The history of recent states are:
Position: **[recent_agent_pos]**, Rotation: **[recent_agent_rotation]**, Action: **[recent_action]**
. . .
Position: **[recent_agent_pos]**, Rotation: **[recent_agent_rotation]**, Current View: **[recent_agent_image]**, Action: **[recent_action]**

---

**4. Prediction Steps:** Please generate the next step given the above states with the following steps: 1) Consider your rotation and position. 2) Check the images to see obstacles or the target object. 3) Decide the action.

---

# B   NATVLM INSTRUCTION DATA

In this section, we give the concrete prompts used in fine-tuning the LVLM, Idefics2.

## B.1   INSTRUCTION TEMPLATE

We show the instruction template we used in the Table 9. There are four parts in the instruction template, including a brief task introduction, episode-specific information, the status of recent steps, and the prediction steps that need to be considered. We leave a lot of placeholders for the episode and step information.

## B.2   RESPONSE TEMPLATE

Previous works usually collect explanation traces using GPT-4 (Mitra et al., 2023; Mukherjee et al., 2023; Ho et al., 2023). In contrast, we collect explanation traces with manually written templates. The templates and examples are shown in Tables 10 and 12. For ROTATERIGHT and ROTATELEFT, we identified the three most common scenarios for rotation based on heuristic rules. Then, we wrote the template for each of them. The first scenario is that the distance difference between the agent and the target object becomes zero in the agent's rotation. Thus, the agent needs to navigate in the other direction. The second scenario involves the presence of obstacles in the agent's current path, necessitating a rotation to navigate around them. The final scenario involves adjusting the agent's rotation to center the target within its field of view, occurring at the end of the navigation process.

Table 10: Response templates we used to build CoT explanation traces for MOVEAHEAD and DONE.

| MOVEAHEAD |
| --- |
| Template:
1) In the direction of my rotation, **[agent_rotation]** degrees (**[cardinal_direction]**), the difference to the target object is **[position_diff]** m. I need to move further **[cardinal_direction]**.
2) There is no obstacle in front of me in recent images.
3) MoveAhead |
| Example:
1) In the direction of my rotation, 90 degrees (east), the difference to the target object is 0.5m. I need to move further east.
2) There is no obstacle in front of me in recent images.
3) MoveAhead |

| DONE |
| --- |
| Template:
1) My position and rotation are equal to the recommended one.
2) I can see the target **[obj_type]** in the image of the current state.
3) Done |
| Example:
1) My position and rotation are equal to the recommended one.
2) I can see the target label marker in the image of the current state.
3) Done |

Table 11: The distribution of collected actions before and after post-processing. The original dataset is very imbalanced since most of the actions are MOVEAHEAD. Then, we downsample the MOVEAHEAD actions with a rate of 0.25. We also filtered the conflicting data.

| Action | w/o Postproc | | w/ Postproc | |
| --- | --- | --- | --- | --- |
| | # Num | % Prop | # Num | % Prop |
| MOVEAHEAD | 221,598 | 77.94% | 57,760 | 49.32% |
| ROTATELEFT | 19,596 | 6.89% | 18,412 | 15.72% |
| ROTATERIGHT | 19,527 | 6.87% | 18,403 | 15.72% |
| DONE | 23,610 | 8.30% | 22,529 | 19.24% |

### B.3 DATA POSTPROCESSING

Directly using all actions in every trajectory to conduct imitation learning brings about an extremely imbalanced dataset. As shown in Table 11, 77.94% actions are MOVEAHEAD. Then we downsampled MOVEAHEAD actions in the instruction dataset. We only retain 25% of the MOVEAHEAD actions resulting in a more balanced dataset.

Then, we also remove conflicting data. We find that steps from different trajectories within the same house occasionally exhibit conflicting information. They have the exact same input information but different action predictions. This happens when two overlapped trajectories diverge at some point due to different target objects. Those conflicting data can confuse the fine-tuned LVLM and lead to worse performance. Thus, we remove those conflicting data from our dataset.

We show the final distribution of our dataset in Table 11 (w/ Postproc).

## C SUPPLEMENTARY EXPERIMENT DETAILS

In this section, we provide supplementary experiment results and show the prompt details.

### C.1 ABLATION STUDY PROMPT

We change the prompt templates for tuning our agent NATVLM in the ablation studies.

Table 12: Response templates we used to build CoT explanation traces for three common rotation scenarios.

| **First scenario: distance difference becomes zero** |
|---|
| Template:
1) In the direction of my rotation, **[agent_rotation]** degrees (**[cardinal_direction]**), the difference to the recommended position is 0.00m. Thus, I need to move in another direction, where the difference is **[other_position_diff]** m, and the rotation is **[other_agent_rotation]** degrees.
2) Obstacles don't affect rotation.
3) RotateRight/RotateLeft |
| Example:
1) In the direction of my rotation, 180 degrees (south), the difference to the recommended position is 0.00m. Thus, I need to move in another direction, where the difference is 1.25m, and the rotation is 90 degrees.
2) Obstacles don't affect rotation.
3) RotateLeft |

| **Second scenario: Obstacles** |
|---|
| Template:
1) In the direction of my rotation, **[agent_rotation]** degrees (**[cardinal_direction]**), the difference compared to the target object is **[position_diff]** m.
2) There are obstacles in front of me, as shown in current images. I need to rotate in another direction. In the other direction, the difference is **[other_position_diff]** m, and the rotation is **[other_agent_rotation]** degrees.
3) RotateRight/RotateLeft |
| Example:
1) In the direction of my rotation, 180 degrees (south), the difference compared to the target object is 1.50m.
2) There are obstacles in front of me, as shown in current images. I need to rotate in another direction. In the other direction, the difference is 1.25m, and the rotation is 270 degrees.
3) RotateRight |

| **Third scenario: View Adjustion** |
|---|
| Template:
1) My position is the same as the recommended one: **[grid_obj_pos]**. However, my rotation is **[agent_rotation]** degrees, facing **[cardinal_direction]**. I need to adjust the rotation to center the target within its field of view.
2) Obstacles don't affect rotation.
3) RotateRight/RotateLeft |
| Example:
1) My position is the same as the recommended one: (0.50, 1.25). However, my rotation is 90 degrees, facing east. I need to adjust the rotation to center the target within its field of view.
2) Obstacles don't affect rotation.
3) RotateRight |

(1) For adding the gold label of position difference ($\diamond$ w/o ET & w Gold and $\diamond$ w Gold Label), we append a new sentence "The difference to the target object is **[position_diff]**" to the end of the **Episode-Specific Information** part of the instruction template.

(2) For removing the explanation traces, the **Prediction Steps** part of the instruction template is replaced with one shorter sentence, "Please generate the next step given the above states."

(3) For the **difference-equation prompt** in the $\diamond$ w Diff-EQ experiment, we replace **[position_diff]** in all explanation traces with the equation: **[grid_obj_pos]** - **[agent_pos]** = **[position_diff]**.

## C.2 FULL RESULTS OF THE IMAGE NUMBER EXPERIMENT

We provide the full results of the image number experiment of all metrics in Table 15.

Table 13: Hyperparameter investigation. We provide different numbers of recent actions and positions to the embodied agent. We use the **difference-equation prompt** in this table. The best performance is bolded.

| Number | Valid | | | Test | | | All | | |
|---|---|---|---|---|---|---|---|---|---|
| | SR | SPL | SEL | SR | SPL | SEL | SR | SPL | SEL |
| 4 steps | 45.37 | 37.76 | 40.04 | 52.47 | 42.87 | 46.45 | 48.92 | 40.31 | 43.25 |
| 8 steps | 54.63 | 45.02 | 46.48 | 54.94 | 44.04 | 47.40 | 54.78 | 44.53 | 46.94 |
| 12 steps | 51.85 | 42.80 | 44.06 | **60.19** | **48.10** | **51.63** | 56.02 | 45.45 | 47.84 |
| 16 steps | **56.48** | **46.39** | **47.89** | 59.26 | 47.52 | 50.82 | **57.87** | **46.95** | **49.36** |

Table 14: The evaluation of the few-shot learning ability of our agent NATVLM. We test our agent with different proportions of sampled episodes in each room. We compare the results with the GPT-4o and test the **difference-equation prompt** in this table.

| % Data | Valid | | | Test | | | Test Diff w GPT-4o | | |
|---|---|---|---|---|---|---|---|---|---|
| | SR | SPL | SEL | SR | SPL | SEL | $\Delta_{SR}$ | $\Delta_{SPL}$ | $\Delta_{SEL}$ |
| 20 | 37.96 | 32.01 | 36.72 | 32.10 | 26.45 | 31.15 | ↓6.17 | ↓5.29 | ↑3.23 |
| 40 | 38.89 | 32.76 | 38.14 | 33.33 | 27.07 | 31.94 | ↓4.94 | ↓4.67 | ↑4.02 |
| 60 | **62.96** | **51.47** | **52.19** | **58.95** | **47.34** | **49.53** | ↑20.68 | ↑15.60 | ↑21.61 |
| 80 | 57.41 | 46.79 | 45.73 | 57.10 | 45.98 | 46.58 | ↑18.83 | ↑14.24 | ↑18.66 |
| 100 | 54.63 | 45.02 | 46.48 | 54.94 | 44.04 | 47.40 | ↑16.67 | ↑12.30 | ↑19.48 |

Table 15: The investigation of the hyperparameter: image number. We provide different numbers of recent visual observations of the embodied agent. Besides the default prompt we use, we also evaluate the difference-equation prompt. We bold the best performance.

(a) Performance of the default prompt.

| Number | Valid | | | Test | | | All | | |
|---|---|---|---|---|---|---|---|---|---|
| | SR | SPL | SEL | SR | SPL | SEL | SR | SPL | SEL |
| 2 images | 50.00 | 41.64 | 40.23 | 50.00 | 40.71 | 41.03 | 50.00 | 41.17 | 40.63 |
| 4 images | **57.41** | **47.84** | **47.90** | 54.94 | 44.45 | 45.83 | **56.17** | **46.15** | **46.86** |
| 6 images | 45.37 | 37.61 | 35.37 | 50.31 | 40.88 | 38.70 | 47.84 | 39.25 | 37.03 |
| 8 images | 49.07 | 40.82 | 41.52 | 54.01 | 43.61 | **46.22** | 51.54 | 42.22 | 43.87 |

(b) Performance of the difference-equation prompt.

| Number | Valid | | | Test | | | All | | |
|---|---|---|---|---|---|---|---|---|---|
| | SR | SPL | SEL | SR | SPL | SEL | SR | SPL | SEL |
| 2 images | 46.30 | 38.54 | 38.93 | 52.16 | 42.13 | 44.76 | 49.23 | 40.34 | 41.84 |
| 4 images | 54.63 | 45.02 | **46.48** | 54.94 | 44.04 | **47.40** | 54.78 | 44.53 | **46.94** |
| 6 images | 49.07 | 40.82 | 38.88 | 51.23 | 41.56 | 40.20 | 50.15 | 41.19 | 39.54 |
| 8 images | **55.56** | **45.81** | 43.40 | **57.41** | **46.39** | 44.40 | **56.48** | **46.10** | 43.90 |

## C.3 COMPLEMENTARY RESULTS OF THE ACTION NUMBER EXPERIMENT

While we provide the results of the action number experiment with the default prompt in Table 3, we also provide the results with the difference-equation prompt in Table 13. The prompt is used as "◇ w Diff-EQ" in the ablation study.

## C.4 COMPLEMENTARY RESULTS OF THE FEW-SHOT EXPERIMENT

While we provide the results of the few-shot learning with the default prompt in Table 4, we also provide the results with the difference-equation prompt in Table 14. The prompt is used as "◇ w Diff-EQ" in the ablation study.

## D  IMPLEMENTATION DETAILS

We train our agent NATVLM from Idefics 2 on 8 NVIDIA A100 GPUs using the Megation-LM framework (NVIDIA, 2021). All parts of the Idefics 2 are fine-tuned, including the LLM, vision encoder, and modality projector. We load the model in BF16 and fine-tune it for one epoch with the learning rate and batch size of 2e-5 and 64, respectively. The best checkpoint is selected according to the sum of all metrics on the validation set. The image size sampled from the AI2THOR is $300 \times 300$. For the baselines, we use the same instructions as our agent and ask them to predict the next action directly. Similarly, we also provide the history of the recent 8 steps (i.e., actions and agent positions) and the visual observation of the recent 4 steps. The exceptions are blind LLMs and Llava 1.5, which can handle zero and one image, respectively. We access the closed-source LVLMs via the OpenAI API[1] with the specific versions of `gpt-4-vision-preview` and `gpt-4o-2024-08-06`.

## E  MORE HOUSE EXAMPLES

In this section, we present more houses built in our DIVSCENE dataset in Figure 8.

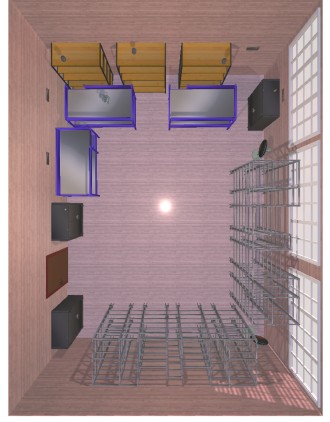

(a) a warehouse with large windows

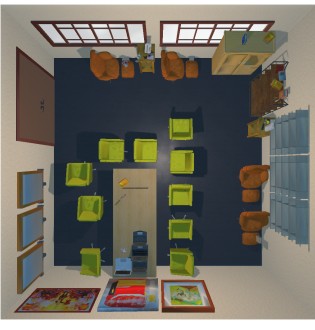

(b) a meeting room with artistic paintings

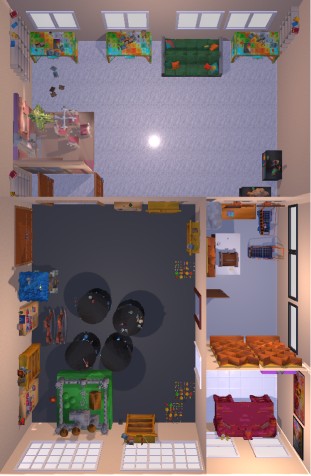

(c) a toy store with a magical kingdom theme.

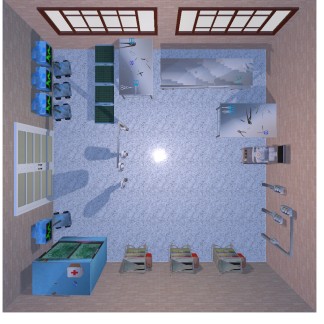

(d) an operating room used by surgeons and nurses featuring light-colored tiles on the walls.

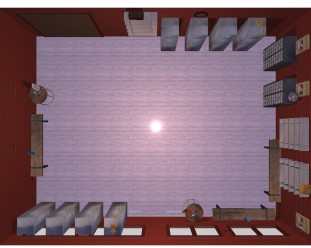

(e) an industrial locker room

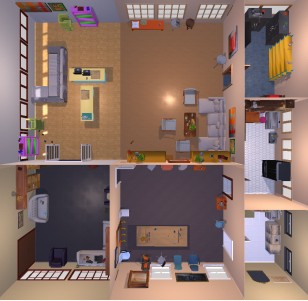

(f) a community center with a versatile and inclusive theme featuring a spacious room size.

Figure 8: Examples of different houses in DIVSCENE.

---

[1]https://platform.openai.com/docs/api-reference

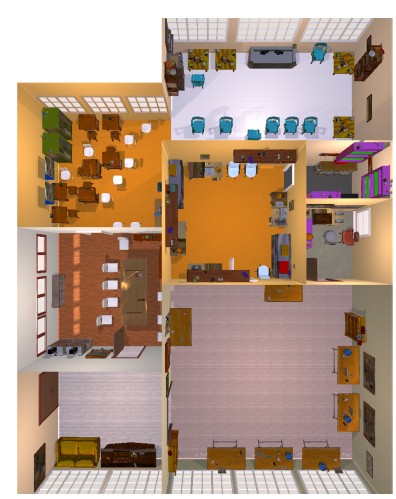
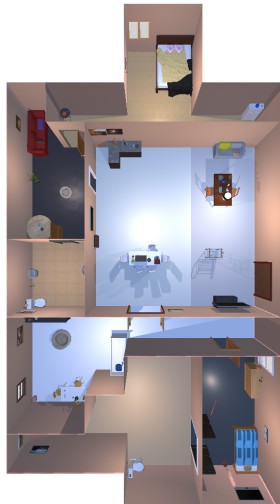

(a) a community center with versatile flooring          (b) a house from ProcTHOR

Figure 9: Comparing our houses with ProcTHOR.

### E.1 COMPARISON WITH PROCTHOR

In Figure 9, we compare houses from our dataset and ProcTHOR with the same number of rooms (8 rooms). Obviously, our scene is more complex with more objects. Quantitively, our scene contains 466, and the scene from ProcTHOR contains only 74 objects.

