# OpenReview forum: "DivScene: Benchmarking LVLMs for Object Navigation with Diverse Scenes and Objects"
_ICLR.cc/2025/Conference — Submitted to ICLR 2025_

### Official Review · Reviewer_AM4v · 2024-10-21

**Soundness:** 2
**Presentation:** 3
**Contribution:** 2
**Rating:** 6
**Confidence:** 5

**Summary:**

This paper presents a new benchmark for object-goal navigation as well as a LVLM-based baseline method. For data generation, this work adopts GPT-4 to generate diverse scene data and samples episodes accordingly. Then NatVLM is proposed as a baseline method, which tunes a LVLM using chain-of-thought. The authors compare NatVLM with other LLM or LVLM navigation methods on the proposed benchmark and demonstrates leading performance.

**Strengths:**

1. This paper is easy to follow.
2. DivScene contains a large amount of houses of different types and object categories, which can serve as a good evaluation benchmark for open-vocabulary object navigaion.

**Weaknesses:**

Although the proposed dataset is diverse, the experimental comparison is weak:
The authors claim in L68-73 "However, they still face a substantial domain gap between navigation tasks and the LLM training corpus (Lin et al., 2024). Moreover, those methods (Chen et al., 2023; Zhu et al., 2024) require a captioning model to generate textual descriptions of each scene, serving as the perception of the backbone LLMs. The captioning model might miss crucial information about objects and lead to suboptimal results, like spatial or color details.". If this is true, the authors should compare with state-of-the-art open-vocabulary object navigation methods [1,2,3] to validate this. Otherwise, only comparing with LLM and LVLMs are less meaningful.

Also, some details about the dataset need to be further clarified:
1. How does the objects arranged in the scenes? How to make sure their relationship is reasonable?
2. How to define scene type and what is the criteria for choosing them?

Moreover, the whole approach, including data generation and NatVLM, is too naive. There are many ways to exploit the ability of LLM/LVLMs for navigation [2,3] and a thorough comparison is required.

Typos:
Line 259: Similarly, we randomly select 27 houses from distinct scene types as the validation set. (validation-->test)

[1] Esc: Exploration with soft commonsense constraints for zero-shot object navigation. ICML 2023.

[2] Vlfm: Vision- language frontier maps for zero-shot semantic navigation. ICRA 2024.

[3] SG-Nav: Online 3D Scene Graph Prompting for LLM-based Zero-shot Object Navigation. NeurIPS 2024.

**Questions:**

See weakness. I hope my questions can be well addressed and more experimental results are required to support the authors' claim.

**Details Of Ethics Concerns:**

This paper proposes a new dataset and benchmark. The authors should make sure they follow the copyright of relevant projects and datasets.

---

> ### Author Response · Authors · 2024-11-21
> **Response to Reviewer AM4v (1/2)**
>
> We thank the reviewer HDKj for their thorough assessment and for recognizing the key contributions of our work: **clear structure**, **large-scale new benchmark**, **a good evaluation benchmark for open-vocabulary object navigation**. Here, we solve your concerns about our work:
>
> > Weakness #1: Although the proposed dataset is diverse, the experimental comparison is weak: The authors claim in L68-73 "However, they still face a substantial domain gap between navigation tasks and the LLM training corpus (Lin et al., 2024). Moreover, those methods (Chen et al., 2023; Zhu et al., 2024) require a captioning model to generate textual descriptions of each scene, serving as the perception of the backbone LLMs. The captioning model might miss crucial information about objects and lead to suboptimal results, like spatial or color details.". If this is true, the authors should compare with state-of-the-art open-vocabulary object navigation methods [1,2,3] to validate this. Otherwise, only comparing with LLM and LVLMs are less meaningful.
>
> To compare those baselines, we added a new experiment on the HM3D dataset [1], which is photorealistic and built on the Habitat platform [2], to support our motivation. Here, we directly test our NatVLM model without further fine-tuning on HM3D to show its zero-shot transferring ability. We compare our methods with various methods: ZSON [4], ESC [5], and ProcTHOR [3]. ZSON and ESC are two zero-shot LLM-based methods. Other than LLM-based methods, ProcTHOR models are smaller models trained using reinforcement learning. We test two ProcTHOR models with and without continual fine-tuning on the HM3D dataset. The results are shown in the table below. We see that our model can also outperform existing methods based on zero-shot LLMs, which strengthens our motivation.
>
> | Model | SPL↑ | SR↑ |
> |-------|------|-----|
> | ProcTHOR | 31.8 | 54.4 |
> | ProcTHOR (zero-shot) | 7.7 | 13.2 |
> | ZSON | 12.6 | 25.5 |
> | ESC | 22.3 | 39.2 |
> |Our|34.11|41.51|
>
> > Weakness #2: Also, some details about the dataset need to be further clarified:
> How does the objects arranged in the scenes? How to make sure their relationship is reasonable? How to define scene type and what is the criteria for choosing them?
>
> We clarify those questions one by one:
> 1) Object arrangement: As we discussed in Section 4.1, we collect scenes based on Holodeck [6] . Specifically, we employ GPT-4 to generate spatial relations between the objects, e.g., “coffee table, in front of, sofa." We treat those GPT-4 generated constraints as soft constraints and also enforce hard constraints to prevent object collisions and ensure that all objects are within the room’s boundaries. Finally, a DFS-based search algorithm is used to find a valid solution to those constraints.
>
> 2) We use LLMs' commonsense knowledge to make sure the relationships between objects and scene types are reasonable. Specifically, for a short house description, like "a dental office with bright lighting," we use GPT-4 to list related objects, such as dental chairs, overhead dental lights, dental instruments, X-ray machines, etc. Thus, we can ensure their relationship. We list a lot of houses in Figures 3 and 8, which can help verify their relationship. You can also check Holodeck [6] for more details.
>
> 3) Our scene types are pre-defined and based on the previous MIT Scenes Dataset [7]. In the dataset, all scenes are classified into 5 big groups, and we manually added more scene types according to those 5 big groups. We have shown a detailed discussion about this in Section 4.1 and Appendix A.
>
> > Weakness #3: Moreover, the whole approach, including data generation and NatVLM, is too naive.
>
> We clarify that Sections 3, 4, and 5 in our paper describe a **novel and advanced framework** for collecting scene data and training LVLMs to finish the navigation task. We train the NatVLM through an end-to-end method by imitating the shortest paths with CoT traces. Also, for the data collection process, we designed the first pipeline to collect houses with diverse (81) scene types and used a BFS-based planner to collect trajectories. To collect different scenes, we use GPT-4 to automatically collect short scene descriptions and build scenes on AI2THOR [8] with Holodeck and GPT-4. We sincerely appreciate more details about the reason why our method is naive.
>
> > Weakness #4:  There are many ways to exploit the ability of LLM/LVLMs for navigation [2,3] and a thorough comparison is required.
>
> As we discussed in Weakness #1, we also tested our method on the HM3D dataset [1], which is photorealistic and built on another platform: Habitat. We also include various baselines, including RL-based ones and zero-shot LLM-based ones. In our paper, we conduct comprehensive analyses in Sections 6.1-6.8. Thus, our experiments provide a thorough comparison.

---

> ### Author Response · Authors · 2024-11-21
> **Response to Reviewer AM4v (2/2)**
>
> # Typo
> Thanks for pointing it out. Our dataset is divided into training, validation, and test sets, covering 4506, 27, and 81 houses. We fixed it in our revised version in line 260.
>
> # Reference
> 1. Ramakrishnan, Santhosh K., et al. "Habitat-matterport 3d dataset (hm3d): 1000 large-scale 3d environments for embodied ai." arXiv preprint arXiv:2109.08238 (2021).
> 2. Savva, Manolis, et al. "Habitat: A platform for embodied ai research." Proceedings of the IEEE/CVF international conference on computer vision. 2019.
> 3. Deitke, Matt, et al. "ProcTHOR: Large-Scale Embodied AI Using Procedural Generation." Advances in Neural Information Processing Systems 35 (2022): 5982-599
> 4. Majumdar, Arjun, et al. "Zson: Zero-shot object-goal navigation using multimodal goal embeddings." Advances in Neural Information Processing Systems 35 (2022): 32340-32352.
> 5. Zhou, Kaiwen, et al. "Esc: Exploration with soft commonsense constraints for zero-shot object navigation." International Conference on Machine Learning. PMLR, 2023.
> 6. Yang, Yue, et al. "Holodeck: Language guided generation of 3d embodied ai environments." Proceedings of the IEEE/CVF Conference on Computer Vision and Pattern Recognition. 2024.
> 7. A. Quattoni, and A.Torralba. Recognizing Indoor Scenes. IEEE Conference on Computer Vision and Pattern Recognition (CVPR), 2009.
> 8. Kolve, Eric, et al. "Ai2-thor: An interactive 3d environment for visual ai." arXiv preprint arXiv:1712.05474 (2017).

---

> > ### Comment · Reviewer_AM4v · 2024-11-24
> >
> > Thanks for the authors' quick reply. It solves some of my concern. However, the experiments in this paper is still too weak to support the claim. As I mentioned, all state-of-the-art LLM-based object-goal navigation methods should be discussed and compared. In the rebuttal, the authors only compare with ZSON and ESC, which are methods about two years ago. SOTA methods like [1][2][3] should also be compared.
> >
> >
> > [1] Vlfm: Vision- language frontier maps for zero-shot semantic navigation. ICRA 2024.
> >
> > [2] InstructNav: Zero-shot System for Generic Instruction Navigation in Unexplored Environment. CoRL 2024.
> >
> > [3] SG-Nav: Online 3D Scene Graph Prompting for LLM-based Zero-shot Object Navigation. NeurIPS 2024.

---

> ### Author Response · Authors · 2024-11-24
> **Response to Reviewer AM4v**
>
> Thanks for your prompt response. We augmented our experiment according to your response. First, we want to clarify that ESC was published in 2023, which is not two years ago. Then, we find that the third paper SG-Nav is arXiv-ed after the deadline of ICLR. Thus, even though we included it in our new experiment below, asking to include this paper is **a violation of the review guideline** (https://iclr.cc/Conferences/2025/ReviewerGuide).
>
> In the augmented experiment below, we test a lot of models on the HM3D dataset. Here, we consider two lines of work: fine-tuned methods and zero-shot methods. For the fine-tuned methods, we list the performance of ProcTHOR [1]  and PIRLNav [2]. ProcTHOR is trained using RL, and PIRLNav is trained using imitation learning. Then, for the zero-shot method, we tested a lot of LLM or VLM-based navigation methods, including ZSON [3], ESC [4], VLFM [5], InstructNav [6], SG-Nav [7]. We can find that our method can achieve a higher SPL score, and previous LLM or VLM-based methods achieve better SR. We want to emphasize that those LLM or VLM-based methods usually involve a lot of extra information than ours. For example, our method only takes ego-centric RGB image as input while those methods [3,4,5,6,7] also get depth information, scene graphs, local policy navigation tools, extra VLMs (like GPT-4) besides backbone VLM, semantic segmentation, panoramic field of views. It is reasonable for them to get a higher SR score.
>
> | Model | SPL↑ | SR↑ |
> |-------|------|-----|
> | **Fine-tuned**      |        |        |
> | ProcTHOR | 31.8 | 54.4 |
> | PIRLNav | 27.1 |  64.1 |
> | **Zero-Shot** |        |        |
> | ProcTHOR (zero-shot) | 7.7 | 13.2 |
> | ZSON | 12.6 | 25.5 |
> | ESC | 22.3 | 39.2 |
> |VLFM| 30.4 | 52.5 |
> |SG-Nav| 24.8 | 53.9 |
> |InstructNav| 20.9 | **58.0** |
> |Our|**34.11**|41.51|
>
>
> # Reference
> 1. Deitke, Matt, et al. "ProcTHOR: Large-Scale Embodied AI Using Procedural Generation." Advances in Neural Information Processing Systems 35 (2022): 5982-599
> 2. Ramrakhya, Ram, et al. "Pirlnav: Pretraining with imitation and rl finetuning for objectnav." Proceedings of the IEEE/CVF Conference on Computer Vision and Pattern Recognition. 2023.
> 3. Majumdar, Arjun, et al. "Zson: Zero-shot object-goal navigation using multimodal goal embeddings." Advances in Neural Information Processing Systems 35 (2022): 32340-32352.
> 4. Zhou, Kaiwen, et al. "Esc: Exploration with soft commonsense constraints for zero-shot object navigation." International Conference on Machine Learning. PMLR, 2023.
> 5. Yokoyama, Naoki, et al. "Vlfm: Vision-language frontier maps for zero-shot semantic navigation." 2024 IEEE International Conference on Robotics and Automation (ICRA). IEEE, 2024.
> 6. Yin, Hang, et al. "SG-Nav: Online 3D Scene Graph Prompting for LLM-based Zero-shot Object Navigation." arXiv preprint arXiv:2410.08189 (2024).
> 7. Long, Yuxing, et al. "InstructNav: Zero-shot System for Generic Instruction Navigation in Unexplored Environment." arXiv preprint arXiv:2406.04882 (2024).

---

> > ### Comment · Reviewer_AM4v · 2024-11-26
> >
> > Thank you for your rebuttal. Now most of my concerns are solved. I'll raise my score.

---

### Official Review · Reviewer_HDKj · 2024-10-29

**Soundness:** 2
**Presentation:** 2
**Contribution:** 2
**Rating:** 5
**Confidence:** 4

**Summary:**

The paper introduces a new dataset with diverse objects and scenes. The scenes are generated using Holodeck, and the trajectories are collected following the shortest path principle. The dataset contains 4,614 scenes across 81 distinct types. It also provides ground-truth actions along with explanation traces to support Chain-of-Thought (CoT) reasoning. The experimental results demonstrate the effectiveness of these explanation traces in enhancing model performance.

**Strengths:**

1. The paper demonstrates that explanation traces are important for action prediction, providing useful insight for model design.
2. The authors utilize a postprocessing procedure to guarantee the data quality.
3. The authors provide the data examples and the source code of open-sourced API evaluation.

**Weaknesses:**

1. Overclaim. The authors claim that they are the first to build a navigational agent based on LVLMs. However, to my knowledge, NaVid [1] has already introduced a navigation agent using LVLMs and deployed the model in real-world scenarios. In contrast, the current work operates entirely within synthetic scenes and trains its model on a single simulator, which limits the novelty and practical significance. This weakens the overall contribution, as the work does not extend beyond what NaVid achieved.

2. Missing Comparison with Existing Navigation Datasets. The manuscript lacks a proper comparison with existing navigation datasets. It is essential to clarify how this dataset differs from previous datasets in terms of data domain, quantity, and quality. Important details such as the number of scenes, object categories, and trajectory complexity in existing datasets should be explicitly mentioned. Without this information, it is challenging to assess the uniqueness or relevance of the proposed dataset.

3. Limited Technical Contribution. Sections 3, 4, and 5 primarily describe a standard pipeline for collecting navigation trajectories, which follows established methodologies from prior works. The paper should make a clearer distinction regarding its unique contributions in these sections to better articulate the novelty. Without this, the technical advancements of the current work appear minimal.

4. Insufficient Baseline Models and Ablation Study. The experimental section is limited, as the authors train only a single model based on Idefics 2 (8B) and compare it with zero-shot models. This narrow focus prevents a thorough evaluation. Including additional baselines, such as traditional reinforcement learning (RL)-based or imitation learning-based navigation models, would provide a more comprehensive comparison. Furthermore, the influence of different LLM backbones or LVLM pre-trained weights should be studied to offer deeper insights into how these components affect navigation performance.


[1] NaVid: Video-based VLM Plans the Next Step for Vision-and-Language Navigation

**Questions:**

1. Zero-shot transfer experiments are not convincing. I visualize the scenes in iTHOR, ProcTHOR, and AI2-THOR and find their styles are very similar, which means a marginal domain gap in your cross-domain setting. So comparing zero-shot models such as llava and qwen-VL with your fine-tuned models makes no sense. A more valuable comparison should be the in-domain experiments on iTHOR and ProcTHOR with your NATVLM.

2. Insufficient Literature Review on Object Navigation. The manuscript does not adequately cover prior research on object navigation, particularly recent works that explore navigation agents built with Vision-Language Models (VLMs). A comprehensive discussion of related works is essential for positioning this study within the existing literature. The omission of relevant research leaves a gap in understanding how the proposed approach compares to prior VLM-based navigation agents and whether it builds on or diverges from previous methodologies.

[1] VLM-Social-Nav: Socially Aware Robot Navigation through Scoring using Vision-Language Models
[2] NaVid: Video-based VLM Plans the Next Step for Vision-and-Language Navigation

---

> ### Author Response · Authors · 2024-11-21
> **Response to Reviewer HDKj (1/3)**
>
> We appreciate the comprehensive review and thank you for acknowledging our work's efforts: 1) **good model design**, 2) **high data quality**, and 3) **open sourcing**. We would like to explain the weaknesses and misunderstandings below:
>
> # Weakness
>
> > Weakness #1: The authors claim that they are the first to build a navigational agent based on LVLMs. However, to my knowledge, NaVid [1] has already introduced a navigation agent using LVLMs and deployed the model in real-world scenarios. In contrast, the current work operates entirely within synthetic scenes and trains its model on a single simulator, which limits the novelty and practical significance. This weakens the overall contribution, as the work does not extend beyond what NaVid achieved.
>
> We want to clarify that our work focuses on the **object navigation** while **NaVid** [1] tries to solve the **Visionand-Language Navigation (VLN)**. The VLN task provides models with very long and detailed instructions besides target objects for finishing a task, which is **usually hard to obtain** when deploying VLN models in the real world. For example, an instruction in object navigation is as short as "Find an apple in the house." In contrast, for a high-level instruction "Rinse off a mug and place it in the coffee maker [2]," the VLN task also provides detailed steps for models to follow: 1) walk to the coffee maker on the right, 2) pick up the dirty mug from the coffee maker, 3) turn and walk to the sink, 4) wash the mug in the sink, 5) pick up the mug and go back to the coffee maker, and 6) put the clean mug in the coffee maker. In our work, we **do not** provide models with long and detailed instructions, which is more challenging. We **sincerely hope** the reviewer can **distinguish different tasks in embodied AI**.
>
> For the question of synthetic scenes, we add a new experiment on the photorealistic dataset, HM3D, to further demonstrate the novelty, practical significance, and overall contribution. Here, we directly test our NatVLM model without further fine-tuning on HM3D to show its zero-shot transferring ability. We compare our methods with various methods: ZSON [4], ESC [5], and ProcTHOR [3]. ProcTHOR models are trained using reinforcement learning, and we test two ProcTHOR models with and without continual fine-tuning on the HM3D dataset. On the other hand, ZSON and ESC are two zero-shot LLM-based methods. The results are shown in the table below. We see that our model can also outperform existing methods on closed-vocab benchmarks, which strengthens the contribution and practical significance of our study.
>
> | Model | SPL↑ | SR↑ |
> |-------|------|-----|
> | ProcTHOR | 31.8 | 54.4 |
> | ProcTHOR (zero-shot) | 7.7 | 13.2 |
> | ZSON | 12.6 | 25.5 |
> | ESC | 22.3 | 39.2 |
> |Our|34.11|41.51|
>
>
> > Weakness #2: Missing Comparison with Existing Navigation Datasets. The manuscript lacks a proper comparison with existing navigation datasets. It is essential to clarify how this dataset differs from previous datasets in terms of data domain, quantity, and quality. Important details such as the number of scenes, object categories, and trajectory complexity in existing datasets should be explicitly mentioned. Without this information, it is challenging to assess the uniqueness or relevance of the proposed dataset.
>
> Thanks for your suggestion. Here, we add extensive comparisons to show our DivScene dataset's uniqueness and relevance compared to existing datasets:
>
> 1) First, in the abovementioned table in Weakness #1, we test our NatVLM model on the photorealistic dataset: HM3D. Our model exhibits high performance even without fine-tuning on HM3D, showing its strong sim-to-real transferring ability. Such results demonstrate that our dataset can approximate real-world settings with **high quality** and equip trained models with strong sim-to-real ability.
>
> 2) Then, we provide a statistical comparison with iTHOR [6] and ProcTHOR to discuss the scene complexity, as those datasets are also built on the AI2THOR platform [6]. The table below shows a detailed comparison. We can find that our dataset contains more types of objects and scenes and more objects per house. This shows that our houses are **more complex**. Meanwhile, we also list the room numbers and average room sizes, where those datasets don't show a large difference.
>
> | Dataset | # object types | # scene type | # object per house | # object types per house | # rooms per house | average room size (m²)|
> |-------|------|-----|-----|-----|-----|-----|
> |ProcTHOR | 38 | 4 | 35.64 | 16.25 | 3.78 | 25.21 |
> |iTHOR | 116 | 4 | 47.26 | 30.92 | 1 | 34.24 |
> |DivScene | 22696 | 81 | 111.42 | 32.17 | 2.35 | 30.15 |
>
> [See continual response for this weakness in the next comment.]

---

> ### Author Response · Authors · 2024-11-21
> **Response to Reviewer HDKj (2/3)**
>
> [Continuation of the Weakness #2]
>
> 3) Also, we find that the complexity of houses can impact navigation performance: diverse objects and scene types can pose more challenges to navigational agents. In the table below, we show the success rates of NatVLM on ProcTHOR and our DivScene dataset. We copy the success rates of NatVLM with CoT and without CoT below from Tables 2 and 5 in our paper. We can observe that NatVLM (without CoT) shows a significant drop of 4.71 when we move the model from ProcTHOR to DivScene (31.25 -> 26.54). This verifies that diverse scene and object types can pose more challenges to navigation models and demonstrates the difficulty of our sampled trajectories. On the other hand, with the CoT, the performance of NatVLM only shows a small fluctuation, showing that our CoT-based instruction tuning can help models better tackle those challenges. To be more concrete, we also show two houses with the same number of rooms from DivScene and ProcTHOR, respectively. The images are shown in Appendix E.1 and Figure 9 in our revised paper. Obviously, our scene is more complex with more objects. Quantitively, our scene contains 466, and the scene from ProcTHOR contains only 74 objects.
>
> | Models | ProcTHOR | DivScene |
> |-------|------|-----|
> | NatVLM | 53.12 | 54.94 |
> | NatVLM (without CoT) | 31.25 | 26.54 |
>
>
> > Weakness #3: Limited Technical Contribution. Sections 3, 4, and 5 primarily describe **a standard pipeline** for collecting navigation trajectories, which follows established methodologies from prior works. The paper should make a clearer distinction regarding its unique contributions in these sections to better articulate the novelty. Without this, the technical advancements of the current work appear minimal.
>
> We clarify that Sections 3, 4, and 5 describe a novel framework for collecting scene data and training LVLMs to finish the navigation task through **an end-to-end method**. According to our literature review, we are the *first* to train LVLMs in an end-to-end method by **imitating the shortest paths** for the object navigation task. Also, for the data collection process, we designed the *first* pipeline to collect houses with diverse (81) scene types and used a BFS-based planner to collect trajectories. To collect different scenes, we use GPT-4 to automatically collect short scene descriptions and build scenes on AI2THOR with Holodeck and GPT-4. **We sincerely appreciate providing any prior work** that may have proposed similar approaches that make our work **a standard pipeline**, as this would help position our contribution in the broader research context.
>
> > Weakness #4: Insufficient Baseline Models and Ablation Study. **The experimental section is limited**, as the authors train only a single model based on Idefics 2 (8B) and compare it with zero-shot models. This narrow focus prevents a thorough evaluation. Including additional baselines, such as traditional reinforcement learning (RL)-based or imitation learning-based navigation models, would provide a more comprehensive comparison.
>
> We emphasize that we conducted comprehensive experiments, including a lot of analyses. For example, we compared our NatVLM with four kinds of baselines in Section 6.3. Then, we conduct ablation studies to study the effectiveness of CoT, prompt template, etc, in Section 6.4. Moreover, we studied the effects of hyperparameters in Section 6.5 and conducted a few-shot learning study in Section 6.6. We also conducted zero-shot transferring experiments in Section 6.7. We conducted comprehensive experiments to demonstrate the effectiveness of our model.
>
> We also added more kinds of baselines. In new experiments on the HM3D dataset, as we mentioned in Weakness #1, we compared our model with RL-based methods (ProcTHOR and ProcTHOR (zero-shot)) and LLM-based methods (ZSON and ESC).
>
> In short, we believe we have already provided extensive experiments to support our study.
>
> # Questions
>
> > Question #1: Zero-shot transfer experiments are not convincing. I visualize the scenes in iTHOR, ProcTHOR, and AI2-THOR and find their styles are very similar, which means a marginal domain gap in your cross-domain setting. So comparing zero-shot models such as llava and qwen-VL with your fine-tuned models makes no sense. A more valuable comparison should be the in-domain experiments on iTHOR and ProcTHOR with your NATVLM.

---

> ### Author Response · Authors · 2024-11-21
> **Response to Reviewer HDKj (3/3)**
>
> Thanks for your concerns about scene similarity and in-domain comparison. First, we want to clarify that iTHOR and ProcTHOR are datasets on the simulator AI2THOR. Thus, iTHOR and ProcTHOR are not comparable alternatives to AI2THOR, and we cannot juxtapose them.
>
> For the scene similarity, as we discussed in Weakness #1, we also included the experiment on the HM3D dataset. HM3D is photorealistic and is on another simulation platform: Habitat. For the in-domain comparison, we also show the performance NatVLM without CoT traces in Table 5 (NATVLM ⋄ w/o ET), which is also fine-tuned on our DivScene dataset. Further, we include the performance of a few LVLMs (like Llava, Qwen-VL, etc) to show the domain gap between the pre-training data of LVLMs and navigation data. This is also one motivation of our model design, which we mentioned in the Introduction (line 068-070).
>
> > Question #2: Insufficient Literature Review on Object Navigation. The manuscript does not adequately cover prior research on object navigation, particularly recent works that explore navigation agents built with Vision-Language Models (VLMs). A comprehensive discussion of related works is essential for positioning this study within the existing literature. The omission of relevant research leaves a gap in understanding how the proposed approach compares to prior VLM-based navigation agents and whether it builds on or diverges from previous methodologies.
>
> Thanks for your recommendation. We will include those papers in our final version. At the same time, we want to emphasize that we have already included sufficient literature in the object navigation task, which is the main focus of our work. As we discussed in Weakness #1, the NaVid paper focuses on the vision language navigation task. Similarly, the VLM-Social-Nav paper focuses on social navigation, which is also different from our topic. We have already discussed the various navigation tasks in lines 106-111. We **should not mix up** those distinct embodied tasks.
>
> # Reference
> 1. Zhang, Jiazhao, et al. "Navid: Video-based vlm plans the next step for vision-and-language navigation." arXiv preprint arXiv:2402.15852 (2024).
> 2. Shridhar, Mohit, et al. "Alfred: A benchmark for interpreting grounded instructions for everyday tasks." Proceedings of the IEEE/CVF conference on computer vision and pattern recognition. 2020.
> 3. Deitke, Matt, et al. "ProcTHOR: Large-Scale Embodied AI Using Procedural Generation." Advances in Neural Information Processing Systems 35 (2022): 5982-599
> 4. Majumdar, Arjun, et al. "Zson: Zero-shot object-goal navigation using multimodal goal embeddings." Advances in Neural Information Processing Systems 35 (2022): 32340-32352.
> 5. Zhou, Kaiwen, et al. "Esc: Exploration with soft commonsense constraints for zero-shot object navigation." International Conference on Machine Learning. PMLR, 2023.
> 6. Kolve, Eric, et al. "Ai2-thor: An interactive 3d environment for visual ai." arXiv preprint arXiv:1712.05474 (2017).

---

> ### Comment · Reviewer_HDKj · 2024-11-25
>
> Thank you for the additional clarifications. The newly included experiments on HM3D, which compare the proposed method with older zero-shot models, address some of my earlier concerns. Since the authors assert that the benchmark is the core contribution of the paper, they should at least analyze how factors such as data scale, the number of object types, and room types impact navigation performance. Unfortunately, the paper currently lacks such a basic analysis. Additionally, the data generation pipeline appears to be straightforward, merely following the standard BFS path sampling approach. I also agree with Reviewer YJpB that the specific combination of the proposed model and benchmark does not sufficiently validate the utility of either the benchmark or the model in isolation.

---

> ### Author Response · Authors · 2024-11-27
> **Response to the new commonse of Reviewer HDKj (1/2)**
>
> We greatly value your **thorough feedback and patience** in reviewing our work. In response to your insightful comments, we have conducted **extensive additional experiments** and provided detailed explanations to address each point raised in your new comment. These improvements have made our paper more comprehensive and rigorous. We sincerely hope our revisions have adequately addressed your concerns and will be grateful if you reconsider these enhancements in your assessment.
>
> # Weaknesses in new comments
>
> > Weakness #1: Since the authors assert that the benchmark is the core contribution of the paper, they should at least analyze how factors such as data scale, the number of object types, and room types impact navigation performance.
>
> First, we already included an experiment to talk about the impact of data scale in the Section 6.6 in our original paper. Our experiment shows that NatVLM can achieve as good as GPT-4 only with 20% data. Also, the performance plateaus when using 80% data. You can check more details in this part.
>
> Then, we also add new experiments of training our model on a different number of scene types. Specifically, we sampled 50% and 80% of our scene types and used the corresponding episodes in houses of those types to train our NatVLM. We find that with more scene types, the model can achieve higher performance with better generalization ability. In contrast, training on 50% gets the worst generalization on our dataset. Given the time limit of the rebuttal period, we will add a full series of tests in the final version of our paper, like the style Section 6.6 in our paper (i.e., including more proportions of 20%, 40%, 60% of scene types, etc) to further augment our paper.
>
> | Model | Valid SPL↑ | Valid SR↑ | Test SPL↑ | Test SR↑ |
> |-------|------------|-----------|-----------|-----------|
> | 50% scene types in our data | 38.78 | 45.47 | 36.13 | 43.97 |
> | 80% scene types in our data | 45.19 | 52.81 | 42.58 | 51.67 |
> | Our | 47.84 | 57.41 | 44.45 | 54.94 |
>
> Then, we also test the effects of diverse objects when training our NatVLM model. First, we train the model to navigate to the 16 target objects listed in ProcTHOR. Then, we use 10% target object types of our DivScene dataset to train the model. We can find the same trends in the above scene types experiments.
>
> | Model | Valid SPL↑ | Valid SR↑ | Test SPL↑ | Test SR↑ |
> |-------|------------|-----------|-----------|-----------|
> | 16 target objects | 21.08 | 24.73| 19.01 | 23.07 |
> | 10% target objects | 32.59 | 37.98 | 30.47 | 36.76 |
> | Our | 47.84 | 57.41 | 44.45 | 54.94 |
>
> > Weakness #2: Additionally, the data generation pipeline appears to be straightforward, merely following the standard BFS path sampling approach.
>
> We want to clarify that the BFS-based planner is only one component of our whole pipeline of data sampling. Specifically, we designed the *first* pipeline to collect houses with diverse (81) scene types and used a BFS-based planner to collect trajectories. To collect different scenes, we use GPT-4 to automatically collect short scene descriptions and build scenes on AI2THOR with Holodeck and GPT-4. Then, we used a BFS-based planner to collect trajectories with diverse target objects. Meanwhile, we want to stress that BFS-based planner is acceptable and also used in other work [10]. Thus, **merely following** is not a correct description of our data collection process.
>
> > Weakness #3: Overall discussion about the combination of method and benchmark: I feel that the current contribution of a combined new method + new benchmark is not yet crisp
>
> Thanks for your suggestion. We believe that with our new experiments, our contribution will be much clearer for the community. **1) For the dataset,** we added a statistical comparison with ProcTHOR and iTHOR, showing that our dataset is more complex and object-cluttered. Meanwhile, we also add experiments of training LVLMs with partial scene types and object types, showing the difficulty of datasets with diverse scenes and objects. **2) For the new model,** we also add new experiments on the HM3D to show the effectiveness of our model, especially its ability to transfer to real-world scenes and cross-platform generalization. **With those new experiments, we believe our paper makes a clear explanation of our contribution to the community.**
>
> Then, for the presenting way you mentioned, we agree with you that presenting the dataset and methods separately like ScanNet is a good way. Meanwhile, we want to stress that combining datasets and methods is also a good alternative to present works. There are a lot of famous works done this way. For example, the ATOMIC paper [8] introduces a multitasking learning method with a hierarchical structure to better model commonsense knowledge. SlideVQA [9] also includes a new model called M3D to solve this problem. Even though different researchers have different tastes in this. We believe we should keep inclusive about writing styles.

---

> ### Author Response · Authors · 2024-11-27
> **Response to the new commonse of Reviewer HDKj (2/2)**
>
> ### In the end, we again appreciate your valuable suggestions and comments. We will be truly grateful if you reconsider your assessment of our work.
>
>
> # Reference
> 1. Deitke, Matt, et al. "ProcTHOR: Large-Scale Embodied AI Using Procedural Generation." Advances in Neural Information Processing Systems 35 (2022): 5982-599
> 2. Ramrakhya, Ram, et al. "Pirlnav: Pretraining with imitation and rl finetuning for objectnav." Proceedings of the IEEE/CVF Conference on Computer Vision and Pattern Recognition. 2023.
> 3. Majumdar, Arjun, et al. "Zson: Zero-shot object-goal navigation using multimodal goal embeddings." Advances in Neural Information Processing Systems 35 (2022): 32340-32352.
> 4. Zhou, Kaiwen, et al. "Esc: Exploration with soft commonsense constraints for zero-shot object navigation." International Conference on Machine Learning. PMLR, 2023.
> 5. Yokoyama, Naoki, et al. "Vlfm: Vision-language frontier maps for zero-shot semantic navigation." 2024 IEEE International Conference on Robotics and Automation (ICRA). IEEE, 2024.
> 6. Yin, Hang, et al. "SG-Nav: Online 3D Scene Graph Prompting for LLM-based Zero-shot Object Navigation." arXiv preprint arXiv:2410.08189 (2024).
> 7. Long, Yuxing, et al. "InstructNav: Zero-shot System for Generic Instruction Navigation in Unexplored Environment." arXiv preprint arXiv:2406.04882 (2024).
> 8. Sap, Maarten, et al. "Atomic: An atlas of machine commonsense for if-then reasoning." Proceedings of the AAAI conference on artificial intelligence. Vol. 33. No. 01. 2019.
> 9. Tanaka, Ryota, et al. "Slidevqa: A dataset for document visual question answering on multiple images." Proceedings of the AAAI Conference on Artificial Intelligence. Vol. 37. No. 11. 2023.
> 10. Ehsani, Kiana, et al. "SPOC: Imitating Shortest Paths in Simulation Enables Effective Navigation and Manipulation in the Real World." Proceedings of the IEEE/CVF Conference on Computer Vision and Pattern Recognition. 2024.

---

> ### Comment · Reviewer_HDKj · 2024-11-27
> **Final decision**
>
> Thank you for the detailed response. The additional experiments suggest that incorporating more objects and scenes consistently enhances performance. While the authors reference other works to support the significance of their contribution, it is essential to critically assess the novelty and impact of these prior efforts.
>
> ATOMIC [1] introduces a groundbreaking task focused on commonsense if-then reasoning, with **a clearly innovative task definition and evaluation pipeline** compared to earlier works. The extensive effort invested in crowdsourcing data annotation and quality evaluation underscores the significance of their contribution.
>
> Similarly, [2] presents a novel VQA benchmark addressing challenges like **multi-image reasoning, multi-hop reasoning, and numerical reasoning**. For the model front, Table 2 showcases **a comprehensive comparison of diverse baseline models (more than 10)**, highlighting the robustness of the proposed benchmark. Alongside defining a new task, the paper introduces a novel framework and conducts **a fair comparison against baseline results**, reflecting substantial innovation and effort.
>
> In contrast, while your work primarily focuses on creating a new benchmark, its task definition and evaluation protocol are the same as those in prior studies. The authors emphasize contributions in expanding the benchmark with more object types, room types, and a minor modification of the BFS algorithm. However, these changes offer very limited novelty compared to the foundational innovations.

---

> ### Author Response · Authors · 2024-11-27
> **Clarification of Our Comprehensive Contributions and Response to the Inappropriate Cross-Domain Comparison**
>
> We appreciate your detailed feedback. However, we would like to clarify that our citation of ATOMIC and SlideVQA was **specifically to illustrate that presenting new methods and datasets within the same paper is a common and accepted practice** in the research community. These examples were not meant to draw direct comparisons in terms of task innovation, as they address **distinctly different research domains**. Our work focuses on embodied AI navigation, which is distinctly different from commonsense reasoning (ATOMIC) or visual question answering (SlideVQA). **Thus, the comparison in your comment digress from the discussion of whether a paper can contain both new datasets and methods**. More importantly, this cross-domain comparison is inappropriate according to the review guideline (https://iclr.cc/Conferences/2025/ReviewerGuide) and misunderstanding our response.
>
> Meanwhile, you need to better understand the contribution and focus of our paper. As we have already talked about similar content twice, we want to stress again that our paper is not only focusing on creating a new benchmark. **1) First**, our paper studies a new task of open-vocabulary object navigation, focusing on generalizable object navigation based on LVLMs. **2) To enable the study**, we designed a novel pipeline to collect a large-scale dataset DivScene with diversified objects and scenes and used a BFS-based planner to collect trajectories. To collect different scenes, we use GPT-4 to automatically collect short scene descriptions and build scenes on AI2THOR with Holodeck and GPT-4. Then, we used a BFS-based planner to collect trajectories with diverse target objects. **3) In this task, we also proposed a new model with CoT traces for better navigation performance.**
>
> More importantly, to support the study in our paper sufficiently, we provided extensive experiments in our paper and added a lot of new experiments in our rebuttal. **1) For the open-vocabulary task and our dataset,** we added a statistical comparison with ProcTHOR and iTHOR, showing that our dataset is more complex and object-cluttered. Meanwhile, we also add experiments of training LVLMs with partial scene types and object types, showing the difficulty of datasets with diverse scenes and objects. **2) For the new model,** we conduct extensive experiments to show its effectiveness from Section 6.1 to Section 6.8. In the rebuttal, we also added a lot of new experiments. We tested our model on the HM3D with a lot of recent baselines to show its effectiveness, especially its ability to transfer to real-world scenes and cross-platform generalization. ​​
>
> We have listed the strong contributions of our work in rebuttal with you twice (and this is the third time), and we hope the reviewer can fairly assess our work without misunderstanding and overlooking content. Other reviewers also recognized the efforts and strengths of our work: 1) open-vocab object navigation is important (Reviewer 441m, YJpB); 2) The method is rigorous with extensive experiments and ablation study (Reviewer 441m, YJpB); 3) The paper is well-organized with logical structure and clear motivation (Reviewer 441m, YJpB, AM4v); 4) The work is very novel (Reviewer YJpB); 5) Important benchmark with good quality (Reviewer AM4v).
>
> **Again, we added a lot of new experiments according to your comments, and we hope you can fairly consider the contribution and efforts of our work.**

---

> ### Comment · Reviewer_HDKj · 2024-11-28
>
> It is disappointing that the authors seem to overlook my important suggestion regarding the experimental section.
> > I also agree with Reviewer YJpB that the specific combination of the proposed model and benchmark does not sufficiently validate the utility of either the benchmark or the model in isolation.
>
> If the novelty of the proposed model and the benchmark are to be emphasized, it is essential to train other methods on your new benchmark to enable a fair comparison and allow readers to objectively assess the method. Simply comparing the results of training-free models, such as VLFM [1] and InstructNav [2], fails to demonstrate the utility of the new benchmark for the object navigation community.
>
> If the authors find it challenging to train models like those in [1] and [2], SG-Nav [3] could serve as a viable substitute. Incorporating such experiments would effectively showcase the value of your benchmark and model design. Specifically:
>
> 1) If SG-Nav [3] achieves enhanced performance on your benchmark compared to its original training set, this would highlight the value of the benchmark.
> 2) Training SG-Nav [3] and comparing its performance with your CoT-based VLM would provide a direct demonstration of your model’s advanced design.
>
>
>
> I have provided a clear and detailed experimental suggestion, including its motivation and procedure. As responsible researchers, it is essential to uphold the principle that **true contributions must be substantiated by robust and convincing experimental results, supported by thorough and detailed analyses, rather than relying solely on claims**. **Similarly, as a responsible reviewer, I am committed to evaluating the true contributions based on solid experimental evidence rather than unverified assertions.**
>
> [1] Yokoyama, Naoki, et al. "Vlfm: Vision-language frontier maps for zero-shot semantic navigation." 2024 IEEE International Conference on Robotics and Automation (ICRA). IEEE, 2024.
>
> [2] Long, Yuxing, et al. "InstructNav: Zero-shot System for Generic Instruction Navigation in Unexplored Environment." arXiv preprint arXiv:2406.04882 (2024).
>
> [3] Yin, Hang, et al. "SG-Nav: Online 3D Scene Graph Prompting for LLM-based Zero-shot Object Navigation." arXiv preprint arXiv:2410.08189 (2024).

---

> > ### Author Response · Authors · 2024-12-02
> > **Final Summary and Thanks for your Comments**
> >
> > Dear the Reviewer HDKj,
> >
> > Thanks for your suggestions. **We want to point out that all three methods (i.e., Vlfm, InstructNav, and SG-Nav) are zero-shot methods. There is no original training set of navigation for us to compare. Then, the main focus of our work is benchmarking current end-to-end LVLMs on the open-vocabulary object navigation, not chasing higher performance on other datasets using our DivScene data.** Basically, our work is not a leaderboard-chasing study on previous object navigation datasets.
> >
> > In our work, we focused on benchmarking the current progress of end-to-end LVLMs on open-vocab object navigation tasks, as our paper title shows. **The true value of our benchmark lies in its diverse objects and scenes for benchmarking the new navigation abilities, not the utility of leader-board chasing on other datasets.** Thus, after building a new benchmark of open-vocab object navigation, the most focused thing is to study how well current LVLMs perform on this challenging task and how to build better end-to-end LVLMs for the task. **The open-vocab object navigation task has not been explored in previous literature and is forward-looking.** That is also why we tested a lot of LVLMs on our benchmark and used CoT traces to build strong LVLMs on the task. (We also show its generalization on the closed-set task.) In our rebuttal, we also benchmarked the performance of those LVLMs trained on the closed-set task. In short, you can see that our experiments are focused on the benchmarking of LVLMs in an end-to-end manner.
> >
> > We admit that your suggestion is excellent (i.e., showing the models trained on our benchmark perform better on closed-set tasks), and we truly appreciate your suggestion. However, the focus on benchmarking has already served as a good alignment between our dataset and methods. **In summary, the motivation for us to build the new dataset is to benchmark how far the current LVLMs are on the open-vocab task, not to show that our dataset is powerful enough to chase better performance in other datasets.** We are not trying to collect a dataset similar to existing ones and chase higher performance with a larger data scale. *(If this were to be true, the motivation would have been "using scaled up diverse data to help closed-set navigation.")*
> >
> > We hope we have explained our motivation and alignment between the dataset and method clearly and you can see that both the benchmark and method are highly aligned with our motivation. We will emphasize and make more clearly about the alignment and the motivation of benchmarking better in our final version.
> >
> > **All in all, we truly appreciate your effort in the discussion.** Your suggestions are really thoughtful and insightful in helping our paper become better.

---

### Official Review · Reviewer_YJpB · 2024-11-02

**Soundness:** 3
**Presentation:** 2
**Contribution:** 2
**Rating:** 6
**Confidence:** 3

**Summary:**

The authors present an open-vocab version of the closed-vocab object navigation. They introduce both a benchmark dataset for this setting, and demonstrate a new VLM-based method (NatVLM).


Benchmark:
* They generate 4k scenes using Holodeck, and aim for diverse generated scenes by using 81 scene types from MIT 365 along with LLMs to generate scene descriptions. They use 22k objects from Objaverse (looks like they subselect 5k somehow).
* They use AI2Thor to compute shortest paths and compute episodes for trianing. Presumably they also use aI2Thor for online evaluation

Baselines:
They present a few baselines, using existing vlms and a method NATVLM  (a finetuned idefics2). They show a few ablations of prompts in the NatVLM baseline. They show zero-shot transfer performance of these baselines on closed-vocab versions ProcThor and iThor.

**Strengths:**

### Significance
Open-vocab version of existing closed-vocab object nav. Reasonably important problem with a large enough community that would be interested in this benchmark.

### Originality
Seems sufficiently novel

### Quality
The authors do a good job of introducing a benchmark using existing tools in the community. They provide results for some reasonable baselines on this new benchmark, and show zero-shot performance on closed-vocab scenes (but still in procthor). Generally they do a very good job describing experimental settings, though some details are missing from the paper -- but would be present in a code release. E.g. which episodes were used for eval on procthor, ithor. The extensive appendix helps here.

### Clarity
Good references to related work, motivation is clear

**Weaknesses:**

My main concern is that the paper does too many things for these experiments to sufficiently support. It reads like a task paper, and a dataset paper, and also a method paper.

1. As a method paper, the submission is missing comparisons to sota on closed-vocab benchmarks (e.g. sota in procthor, or on MP3D in habitat).
2. As a dataset paper: the submission could use more analysis of well these scenes approximate real-world settings -- should we use/trust this data
3. As primarily proposing the task of open-vocab object nav: I would like to see more of a breakdown of where current methods fail, and why this problem is hard

I think there is something here, but it’s not quite ready in the current form. I'll ask/recommend specific experiemnts these in the "questions" section, since I think they could make the submission stronger.

**Questions:**

### As a method:
- In this case the paper would be a method for open-vocab object navigation, where they generate a dataset for the proposed method. But for an open-vocab method, I want to be able to compare its performance to SotA on existing object nav benchmarks, even if they are closed vocab. Only comparisons are to other VLM-based methods, which are not SotA on existing benchmarks.
- You don't need to be the best in closed-vocab objnav. It's okay to break out the open-vocab and closed-vocab results separately in a table. But it would be helpful to show the reader how big the gap is. If there's a large gap, that's an even better motivation of why closed-vocab approaches do not work in real-world settings.
- If you do want to be the best on closed-vocab objnav, you don't need to show zero-shot performance but can finetune/anneal the model on the downstream dataset. Does the more general open-vocab object nav help them improve over sota?

### As a task:
No breakdown on where the failures were. Why is open-vocab much harder than closed-vocab? E.g. how many of the failures were due to missing the object, vs navigation failure. That would allow this to tie into existing literature like 3d object detection (e.g. on scannet, mp3d, etc).

### As a benchmark dataset
I would like to see more analysis of the decisions made for this benchmark, and how well models trained on this data transfer to real-wold settings.
* Existing zero-shot transfer experiments are to other benchmarks also in AI2Thor. So largely this concerns whether the model will work in other ai2thor scenes from procthor/ithor, and using those closed-vocab objects instead of the ones from objaverse
* I would love to see generalization to some obj nav in other simulators and using more real-world data (e.g. ARKit or MP3D, which the authors mention but do not compare to).
* Is generating new scenes Holodeck better than using an existing artist-deigned dataset like Structure3d, which has 3.5k scenes? ARKit which is 4500 real ones and already has a bunch of CAD annotations with cad-estate?
* Is 4k scenes the right number?  Would want to see an experiment when training on subsets of the data to show performance isn't saturating (e.g. 1%, 10%, 50%, 100% of scenes).

---

> ### Author Response · Authors · 2024-11-21
> **Response to Reviewer YJpB (1/3)**
>
> We appreciate your thorough review. Thanks for recognizing our efforts in **focusing on reasonably important problem**, **sufficiently novelty**, **clear motivation**, and **comprehensive details**. We will respond to the weaknesses and concerns below:
>
> # Weakness
>
> > Weakness #1: As a method paper, the submission is missing comparisons to sota on closed-vocab benchmarks (e.g. sota in procthor, or on MP3D in habitat).
>
> Thanks for your suggestions. Here, we add new experiments to the photorealistic dataset HM3D [1] and compare our methods with various methods: ZSON [3], ESC [4], and ProcTHOR [2]. ProcTHOR models are trained using reinforcement learning, and we test two ProcTHOR models with and without continual fine-tuning on the HM3D dataset. On the other hand, ZSON and ESC are two zero-shot LLM-based methods. We directly test our NatVLM model without further fine-tuning on HM3D to test its zero-shot transferring ability. The results are shown in the table below. We see that our model can also outperform existing methods on closed-vocab benchmarks.
>
> | Model | SPL↑ | SR↑ |
> |-------|------|-----|
> | ProcTHOR | 31.8 | 54.4 |
> | ProcTHOR (zero-shot) | 7.7 | 13.2 |
> | ZSON | 12.6 | 25.5 |
> | ESC | 22.3 | 39.2 |
> |Our|34.11|41.51|
>
> > Weakness #2: As a dataset paper: the submission could use more analysis of well these scenes approximate real-world settings -- should we use/trust this data.
>
> We conduct extensive comparisons to show that our DivScene dataset is trustworthy. First, in the abovementioned table, we test our NatVLM model on the photorealistic dataset: HM3D. Our model exhibits high performance even without fine-tuning on HM3D, showing its strong sim-to-real transferring ability. Such results demonstrate that our dataset can approximate real-world settings and equip trained models with strong sim-to-real ability. This is due to GPT-4's strong commonsense ability when building those houses in our datasets.
>
> Then, we provide a statistical comparison with iTHOR [5] and ProcTHOR [2] to discuss the scene complexity, as those datasets are also built on the AI2THOR platform [5]. The table below shows a detailed comparison. We can find that our dataset contains more types of objects and scenes and more objects per house. This shows that our houses are more cluttered and complex. Meanwhile, we also list the room numbers and average room sizes, where those datasets don't show a large difference.
>
> | Dataset | # object types | # scene type | # object per house | # object types per house | # rooms per house | average room size (m²)|
> |-------|------|-----|-----|-----|-----|-----|
> |ProcTHOR | 38 | 4 | 35.64 | 16.25 | 3.78 | 25.21 |
> |iTHOR | 116 | 4 | 47.26 | 30.92 | 1 | 34.24 |
> |DivScene | 22696 | 81 | 111.42 | 32.17 | 2.35 | 30.15 |
>
> Also, we find that the complexity of houses can impact navigation performance: diverse objects and scene types can pose more challenges to navigational agents. In the table below, we show the success rates of NatVLM on ProcTHOR and our DivScene dataset. We copy the success rates of NatVLM with CoT and without CoT below from Tables 2 and 5 in our paper. We can observe that NatVLM (without CoT) shows a significant drop of 4.71 when we move the model from ProcTHOR to DivScene (31.25 -> 26.54).
>
> | Models | ProcTHOR | DivScene |
> |-------|------|-----|
> | NatVLM | 53.12 | 54.94 |
> | NatVLM (without CoT) | 31.25 | 26.54 |
>
> To be more concrete, we also show two houses with the same number of rooms from DivScene and ProcTHOR, respectively. The images are shown in Appendix E.1 and Figure 9 in our revised paper. Obviously, our scene is more complex with more objects. Quantitively, our scene contains 466, and the scene from ProcTHOR contains only 74 objects.
>
> > Weakness #3: As primarily proposing the task of open-vocab object nav: I would like to see more of a breakdown of where current methods fail, and why this problem is hard
>
> Open-vocab object navigation requires the navigational agent to identify thousands of different target objects. Thus, it poses more challenges than current closed-vocab ones.
> Here, we take ProcTHOR as an example. In ProcTHOR, there are only 16 types of target objects: Alarm Clock, Apple, Baseball Bat, Basketball, Bed, Bowl, Chair, Garbage Can, House Plant, Laptop, Mug, Sofa, Spray Bottle, Television, Toilet, and Vase. This set is too small even to cover common objects, like microwave, refrigerator, desk, table, mirror, and bookshelf. This limited object coverage significantly restricts the model's real-world applicability. In contrast, our open-vocab data contain 5707 different types of target objects, such as mixing console stand, modern side table, vintage wooden bench, multicolored bookshelf, etc.
>
> [See continual response for this weakness in the next comment.]

---

> ### Author Response · Authors · 2024-11-21
> **Response to Reviewer YJpB (2/3)**
>
> [Continuation of the Weakness #3]
> Concretely, we use MPNET [6] to convert those types of the two sets into embeddings and compute the cosine similarity between them. For each object type in our dataset, we calculated its similarity with all 16 types in ProcTHOR and selected the maximum. Only 3% of our object types achieved a maximum similarity above 0.7, showing the diversity of target objects and the difficulty of our task. If we only train a model on those 16 object types, it will fail in most cases in our dataset.
>
>
> # Question
>
> > Question #1: In this case the paper would be a method for open-vocab object navigation, where they generate a dataset for the proposed method. But for an open-vocab method, I want to be able to compare its performance to SotA on existing object nav benchmarks, even if they are closed vocab. Only comparisons are to other VLM-based methods, which are not SotA on existing benchmarks.
>
> As we discussed in Weakness #1, we tested our NatVLM model on the existing benchmark, HM3D, with a few SoTA models, including ZSON, ESC, and ProcTHOR models. Please see more details of the experiment in the response to Weakness #1. Our model can also achieve high performance on the existing closed-vocab benchmark.
>
> > Question #2: I would like to see more analysis of the decisions made for this benchmark, and how well models trained on this data transfer to real-wold settings. 1) Existing zero-shot transfer experiments are to other benchmarks also in AI2Thor. So largely this concerns whether the model will work in other ai2thor scenes from procthor/ithor, and using those closed-vocab objects instead of the ones from objaverse 2) I would love to see generalization to some obj nav in other simulators and using more real-world data (e.g. ARKit or MP3D, which the authors mention but do not compare to). 3) Is generating new scenes Holodeck better than using an existing artist-deigned dataset like Structure3d, which has 3.5k scenes? ARKit which is 4500 real ones and already has a bunch of CAD annotations with cad-estate? 4) Is 4k scenes the right number? Would want to see an experiment when training on subsets of the data to show performance isn't saturating (e.g. 1%, 10%, 50%, 100% of scenes).
>
> We solve the concerns in this question one by one:
> 1) As we discussed in Weakness #1, we tested our model on the HM3D dataset, which is a photorealistic (real-world) datasheet on the Habitat platform [7] (not AI2THOR). The experiment shows that our model can achieve high performance.
> 2) Same as above.
> 3) For the comparison of different scene datasets, we provide a comprehensive comparison of our dataset with ProcTHOR and iTHOR in Weakness #2. As the review mentioned, there are also many real-world or artist-designed datasets, like HM3D [1], MP3D [8], ScanNet [9], and Structure3D [10]. However, there are a few drawbacks to those datasets: 1. those datasets still focus on limited scene types. For example, structure3D and HM3D mainly focus on personal living spaces, including living rooms, kitchens, bedrooms, and bathrooms. 2. Those scanned datasets usually cannot support interaction, which can be a future extension of our study. We can train models to not only navigate to open-vocab objects but also interact with them. 3. Those datasets are very labor-intensive and require a lot of scanning or designing work. Our dataset and study show that using an LLM can also provide a large-scale scene dataset that can scale up easily.
> 4) For the question of whether 4.6K scenes are enough, we want to clarify that we have already included an analysis using fewer data in Table 4 and Section 6.6. Our experiment shows that NatVLM can achieve as good as GPT-4 only with 20% data. Also, the performance plateaus when using 80% data. You can check more details in those parts.

---

> > ### Author Response · Authors · 2024-11-21
> > **Response to Reviewer YJpB (3/3)**
> >
> > # Reference:
> > 1. Ramakrishnan, Santhosh K., et al. "Habitat-matterport 3d dataset (hm3d): 1000 large-scale 3d environments for embodied ai." arXiv preprint arXiv:2109.08238 (2021).
> > 2. Deitke, Matt, et al. "ProcTHOR: Large-Scale Embodied AI Using Procedural Generation." Advances in Neural Information Processing Systems 35 (2022): 5982-599
> > 3. Majumdar, Arjun, et al. "Zson: Zero-shot object-goal navigation using multimodal goal embeddings." Advances in Neural Information Processing Systems 35 (2022): 32340-32352.
> > 4. Zhou, Kaiwen, et al. "Esc: Exploration with soft commonsense constraints for zero-shot object navigation." International Conference on Machine Learning. PMLR, 2023.
> > 5. Kolve, Eric, et al. "Ai2-thor: An interactive 3d environment for visual ai." arXiv preprint arXiv:1712.05474 (2017).
> > 6. Song, Kaitao, et al. "Mpnet: Masked and permuted pre-training for language understanding." Advances in neural information processing systems 33 (2020): 16857-16867.
> > (note: we use the sentence bert version: https://huggingface.co/sentence-transformers/all-mpnet-base-v2)
> > 7. Savva, Manolis, et al. "Habitat: A platform for embodied ai research." Proceedings of the IEEE/CVF international conference on computer vision. 2019.
> > 8. Chang, Angel, et al. "Matterport3d: Learning from rgb-d data in indoor environments." arXiv preprint arXiv:1709.06158 (2017).
> > 9. Dai, Angela, et al. "Scannet: Richly-annotated 3d reconstructions of indoor scenes." Proceedings of the IEEE conference on computer vision and pattern recognition. 2017.
> > 10. Zheng, Jia, et al. "Structured3d: A large photo-realistic dataset for structured 3d modeling." Computer Vision–ECCV 2020: 16th European Conference, Glasgow, UK, August 23–28, 2020, Proceedings, Part IX 16. Springer International Publishing, 2020.

---

> ### Comment · Reviewer_YJpB · 2024-11-25
>
> **I appreciate the discussion and additional experiments**
>
> Thank you to the authors for the thoughtful replies. I especially appreciate the additional experiments on HM3D, and the experiments using additional baselines. I respect the work and rigor that goes into these.
>
> The new approach for the LVLM agent does a little bit better along some metrics (SPL) than existing approaches, but worse along success rate (which is the more important of the two, if you have to pick one!). These new baselines also seem significantly stronger than the ones used in the main paper -- though it's hard to tell since they were evaluated on different benchmarks. Still, since the importance of this paper is mainly in the benchmark, I feel that the fact that the experiments are on HM3D makes the additional experiments useful, regardless of the results.
>
> **TLDR**
>
> I am going to keep my current rating, although with the new experiments I considered raising my rating from 5 to 6. I feel that the current contribution of a combined new method + new benchmark is not yet crisp enough for an ICLR paper. Because the proposed agent is mainly being evaluated on a new benchmark, it would be hard for others to gauge the performance outside of the benchmark. On other benchmarks (e.g. HM3D), other methods do comparably -- depending on the metric of choice. While there are some useful experiments that others might benefit from, it would require careful reading of this paper in order to pull out those insights.
>
> **Reasoning**
> It seems that there are several new components here that are more incremental innovations on existing benchmarks, approaches, and additional data sets. I would prefer to see these new components being introduced in separate papers, since I believe that introducing the components separately will make the contribution of each component clearer and each paper easier to read. ScanNet200 does a pretty good job of introducing a benchmark https://rozdavid.github.io/scannet200, IMO.
>
> I think papers are a bit like pull requests in a large codebase.  The big codebase is "community knowledge"; and concise + clean PRs are easier to review and build on than one large PR made of several smaller contributions.
>
>
> **Minor clarification**
> This is more minor, but just a small point of clarification:
> >  we want to clarify that we have already included an analysis using fewer data in Table 4 and Section 6.6. Our experiment shows that NatVLM can achieve as good as GPT-4 only with 20% data
>
> I believe that Table 4 shows a constant number of scenes but varying numbers of episodes per scene (1 - 5 episodes). I am asking about using varying numbers of scenes, but keeping constant the number of episodes per scene. The difference is that I want to understand how important seeing diverse environments for generalization.

---

> ### Author Response · Authors · 2024-11-27
> **Response to the new comments of Reviewer YJpB (1/2)**
>
> Thanks a lot for your prompt reply and patience with our new experiments. We sincerely appreciate your **insightful comments and concerns**. Thus, we have added **more experiments and explanations** to comprehensively address all points raised in your review and make our paper more rigorous and complete. Again, we truly cherish your valuable suggestions and hope we have satisfactorily solved all your concerns. We would be grateful if you would reconsider your assessment of our work.
>
> # Weaknesses in new comments
>
> > Weakness #1: The new approach for the LVLM agent does a little bit better along some metrics (SPL) than existing approaches, but worse along success rate. These new baselines also seem significantly stronger than the ones used in the main paper -- though it's hard to tell since they were evaluated on different benchmarks. Still, since the importance of this paper is mainly in the benchmark, I feel that the fact that the experiments are on HM3D makes the additional experiments useful, regardless of the results.
>
> Thanks for your comment. In our rebuttal, we compared our model with a lot of baselines on the HM3D dataset, including two lines of work: fine-tuned methods and zero-shot methods. As you say, we totally agree that **the experiments on HM3D make the additional experiments useful**. Meanwhile, we want to emphasize that our method achieves not only a little bit better SPL scores. The SPL scores of baselines are mostly lower than 30%. The SPL score of our model is 34.11%, 10 points higher than theirs. Meanwhile, we want to mention that those LLM or VLM-based methods usually involve a lot of extra information than ours. For example, our method only takes ego-centric RGB images as input while those methods also get depth information, scene graphs, local policy navigation tools, extra VLMs (like GPT-4) besides backbone VLM, semantic segmentation, and panoramic field of views.
>
> | Model | SPL↑ | SR↑ |
> |-------|------|-----|
> | **Fine-tuned**      |        |        |
> | ProcTHOR [1] | 31.8 | 54.4 |
> | PIRLNav [2] | 27.1 |  64.1 |
> | **Zero-Shot** |        |        |
> | ProcTHOR (zero-shot) | 7.7 | 13.2 |
> | ZSON [3] | 12.6 | 25.5 |
> | ESC [4] | 22.3 | 39.2 |
> |VLFM [5] | 30.4 | 52.5 |
> |SG-Nav [6] | 24.8 | 53.9 |
> |InstructNav [7] | 20.9 | **58.0** |
> |Our|**34.11**|41.51|
>
>
> > Weakness #2: I am asking about using varying numbers of scenes but keeping the number of episodes per scene constant. The difference is that I want to understand how important seeing diverse environments is for generalization.
>
> Here, we add new experiments of training our model on a different number of scene types. Specifically, we sampled 50% and 80% of our scene types and used the corresponding episodes in houses of those types to train our NatVLM. We find that with more scene types, the model can achieve higher performance with better generalization ability. In contrast, training on 50% gets the worst generalization on our dataset. Given the time limit of the rebuttal period, we will add a full series of tests in the final version of our paper, like the style Section 6.6 in our paper (i.e., including more proportions of 20%, 40%, 60% of scene types, etc) to further augment our paper.
>
> | Model | Valid SPL↑ | Valid SR↑ | Test SPL↑ | Test SR↑ |
>  |-------|------------|-----------|-----------|-----------|
> | 50% scene types in our data | 38.78 | 45.47 | 36.13 | 43.97 |
> | 80% scene types in our data | 45.19 | 52.81 | 42.58 | 51.67 |
> | Our | 47.84 | 57.41 | 44.45 | 54.94 |
>
> Then, we also test the effects of diverse objects when training our NatVLM model. First, we train the model to navigate to the 16 target objects listed in ProcTHOR. Then, we use 10% target object types of DivScene to train NatVLM. We can find the same trends in the above scene types experiments.
>
> | Model | Valid SPL↑ | Valid SR↑ | Test SPL↑ | Test SR↑ |
> |-------|------------|-----------|-----------|-----------|
> | 16 target objects | 21.08 | 24.73| 19.01 | 23.07 |
> | 10% target objects | 32.59 | 37.98 | 30.47 | 36.76 |
> | Our | 47.84 | 57.41 | 44.45 | 54.94 |
>
> > Weakness #3: Overall discussion about the combination of method and benchmark: I feel that the current contribution of a combined new method + new benchmark is not yet crisp
>
> Thanks for your suggestion. We believe that with our new experiments, our contribution will be much clearer for the community. **1) For the dataset,** we added a statistical comparison with ProcTHOR and iTHOR, showing that our dataset is more complex and object-cluttered. Meanwhile, we also add experiments of training LVLMs with partial scene types and object types, showing the difficulty of datasets with diverse scenes and objects. **2) For the new model,** we also add new experiments on the HM3D to show the effectiveness of our model, especially its ability to transfer to real-world scenes and cross-platform generalization. **With those new experiments, we believe our paper makes a clear explanation of our contribution to the community.**

---

> > ### Author Response · Authors · 2024-11-27
> > **Response to the new comments of Reviewer YJpB (2/2)**
> >
> > Then, for the presenting way you mentioned, we agree with you that presenting the dataset and methods separately like ScanNet is a good way. Meanwhile, we want to stress that combining datasets and methods is also a good way to present works. There are a lot of famous works done this way. For example, the ATOMIC paper [8] introduces a multitasking learning method with a hierarchical structure to better model commonsense knowledge. SlideVQA [9] also includes a new model called M3D to solve this problem. Even though different researchers have different tastes in this. We believe we should keep inclusive about this.
> >
> > **In the end, we again appreciate your valuable suggestions and comments. We will be truly grateful if you reconsider your assessment of our work.**
> >
> >
> > # Reference
> > 1. Deitke, Matt, et al. "ProcTHOR: Large-Scale Embodied AI Using Procedural Generation." Advances in Neural Information Processing Systems 35 (2022): 5982-599
> > 2. Ramrakhya, Ram, et al. "Pirlnav: Pretraining with imitation and rl finetuning for objectnav." Proceedings of the IEEE/CVF Conference on Computer Vision and Pattern Recognition. 2023.
> > 3. Majumdar, Arjun, et al. "Zson: Zero-shot object-goal navigation using multimodal goal embeddings." Advances in Neural Information Processing Systems 35 (2022): 32340-32352.
> > 4. Zhou, Kaiwen, et al. "Esc: Exploration with soft commonsense constraints for zero-shot object navigation." International Conference on Machine Learning. PMLR, 2023.
> > 5. Yokoyama, Naoki, et al. "Vlfm: Vision-language frontier maps for zero-shot semantic navigation." 2024 IEEE International Conference on Robotics and Automation (ICRA). IEEE, 2024.
> > 6. Yin, Hang, et al. "SG-Nav: Online 3D Scene Graph Prompting for LLM-based Zero-shot Object Navigation." arXiv preprint arXiv:2410.08189 (2024).
> > 7. Long, Yuxing, et al. "InstructNav: Zero-shot System for Generic Instruction Navigation in Unexplored Environment." arXiv preprint arXiv:2406.04882 (2024).
> > 8. Sap, Maarten, et al. "Atomic: An atlas of machine commonsense for if-then reasoning." Proceedings of the AAAI conference on artificial intelligence. Vol. 33. No. 01. 2019.
> > 9. Tanaka, Ryota, et al. "Slidevqa: A dataset for document visual question answering on multiple images." Proceedings of the AAAI Conference on Artificial Intelligence. Vol. 37. No. 11. 2023.

---

> ### Author Response · Authors · 2024-12-02
> **Final Summary and Thanks for your Comments**
>
> Dear the Reviewer YJpB,
>
> Thanks for your insightful suggestions. During our discussion, we added many new experiments that are supportive of the claim of work and solved all of your concerns except the presentation.
>
> We appreciate your last comments about the presentation. We agree that we can test the LVLMs on existing closed-set object navigation to enrich our experiment. However, in our work, we focused on benchmarking the current progress of end-to-end LVLMs on open-vocab object navigation tasks, as our paper title shows. **The true value of our benchmark lies in its diverse objects and scenes for benchmarking new navigation abilities, not the utility of leader-board chasing on other datasets.** Thus, after building a new benchmark of open-vocab object navigation, the most focused thing is to study how well current LVLMs perform on this challenging task and how to build better end-to-end LVLMs for the task. **The open-vocab object navigation task has not been explored in previous literature and is forward-looking.** That is also why we tested a lot of LVLMs on our benchmark and used CoT traces to build strong LVLMs on the task. (We also show its generalization on the closed-set task.) In our rebuttal, we also benchmarked the performance of those LVLMs trained on the closed-set task. In short, you can see that our experiments are focused on the benchmarking of LVLMs in an end-to-end manner.
>
> We admit that your suggestion is excellent (i.e., showing the models trained on our benchmark perform better on closed-set tasks), and we truly appreciate your suggestion. However, the focus on benchmarking has already served as a good alignment between our dataset and methods. **In summary, the motivation for us to build the new dataset is to benchmark how far the current LVLMs are on the open-vocab task, not to show that our dataset is powerful enough to chase better performance in other datasets.** We are not trying to collect a dataset similar to existing ones and chase higher performance with a larger data scale. *(If this were to be true, the motivation would have been "using scaled up diverse data to help closed-set navigation.")*
>
> We hope we have explained our motivation and alignment between the dataset and method clearly and you can see that both the benchmark and method are highly aligned with our motivation. We will emphasize and make more clearly about the alignment and the motivation of benchmarking better in our final version.
>
> **All in all, we truly appreciate your effort in the discussion.** Your suggestions are really thoughtful and insightful in helping our paper become better.

---

### Official Review · Reviewer_441m · 2024-11-03

**Soundness:** 4
**Presentation:** 3
**Contribution:** 4
**Rating:** 8
**Confidence:** 3

**Summary:**

his paper tackles the challenge of object navigation in varied environments by introducing DIVSCENE, a dataset with over 4,600 scenes across 81 types. Building on this, the authors developed NATVLM, a navigation agent fine-tuned on a large vision-language model. NATVLM uses Chain of Thought reasoning to help it understand navigation steps better, which improves its accuracy and generalization in diverse settings. The experiments show strong performance against other models and confirm the agent's adaptability across different datasets.

**Strengths:**

1. This paper introduces the DIVSCENE dataset, which is a substantial addition to the field of embodied AI. Unlike existing datasets that are often limited to a narrow range of scenes or objects, DIVSCENE covers 81 diverse scene types with over 22,000 object types, enhancing the scope for studying generalizable object navigation.
2. The method is rigorous, with extensive experiments that include comparisons to baseline models (both open-source and closed-source) and ablation studies to evaluate the impact of CoT reasoning. The experiments provide solid evidence that NATVLM significantly outperforms existing methods in diverse navigation scenarios.
3. The paper is well-organized and presents the main ideas in a clear and logical structure.

**Weaknesses:**

1. While DIVSCENE provides a broad range of synthetic environments, the reliance on simulated platforms (like AI2THOR) limits the real-world applicability of NATVLM. Simulated environments often lack the unpredictable variability and physical constraints found in real-world scenarios. Expanding experiments to include real-world datasets or environments would strengthen the evidence for the model’s generalization abilities and applicability. Including a discussion on the challenges of transferring synthetic-trained models to real-world settings would also be beneficial.
2.  The manual creation of CoT explanation traces for training NATVLM may limit scalability and automation in more complex or dynamic environments.
3. While DIVSCENE offers diverse scenes and objects, the paper lacks a detailed analysis of the scene complexity and its impact on navigation performance. For example, understanding how NATVLM performs in cluttered versus minimal scenes, or in spaces with varying levels of visual occlusion, would add depth to the evaluation. Such insights could help clarify the scenarios where NATVLM’s CoT-based reasoning is most beneficial and identify cases where additional model improvements may be needed.
4. Although the paper emphasizes the effectiveness of NATVLM, it does not address the computational cost of training and deploying the model, especially given the integration of large-scale LVLMs and manually designed CoT traces. Providing a comparison of training and inference times with baseline models would offer readers a clearer understanding of the model’s practicality.

**Questions:**

Please refer to the weaknesses section.

---

> ### Author Response · Authors · 2024-11-21
> **Response to Reviewer 441m (1/3)**
>
> We appreciate your detailed feedback. Thanks for recognizing the strengths of our work as **making a substantial contribution to embodied AI**, **enhancing the scope for generalizable object navigation**, **extensive experiments**, and **well-organized and clear structure**. We would like to respond to the weaknesses and concerns:
>
> # Weakness
>
> > Weakness #1: While DIVSCENE provides a broad range of synthetic environments, *the reliance on simulated platforms* (like AI2THOR) limits the real-world applicability of NATVLM. Simulated environments often lack the unpredictable variability and physical constraints found in real-world scenarios. *Expanding experiments to include real-world datasets or environments* would strengthen the evidence for the model’s generalization abilities and applicability. Including a discussion on the challenges of transferring synthetic-trained models to real-world settings would also be beneficial.
>
> First, current simulation platforms are strong enough to provide reliable virtual environments. For the variability, works on AI2THOR [2,3] employ the large-scale object set, Objaverse [4], to ensure the diversity of scenes and GPT-4 to ensure houses are consistent with human commonsense. For physical constraints, current simulation platforms AI2THOR [2,3] and Habitat [5] use physics engines Unity3D and Bullet Physics, respectively, to simulate physical properties. Moreover, there are a lot of simulated datasets that scan real houses to ensure variability and physics, like MP3D [6], HM3D [7], and Gibson [8]. Thus, using current strong simulation platforms can provide reliable experiment results. Meanwhile, simulator-based data collection is necessary for current data collection for rapid iteration and extensive testing. Real-world data collection is prohibitively time-consuming and resource-intensive. For example, RT-1 [1] uses 17 months to collect enough data to train a 19M transformer.
>
> Then, we also include new experiments on real-world datasets and discuss the transferring ability of our methods. Here, we test our NatVLM model on the photorealistic dataset: HM3D, on the Habitat platform. We directly test our model without further fine-tuning on HM3D in a zero-shot manner. We compare our model with a few baselines, including ProcTHOR [9], ZSON [10], ProcTHOR (zero-shot), and ESC [11]. Compared to methods without fine-tuning on HM3D (i.e., ZSON, ESC, and ProcTHOR (zero-shot)), our method can achieve higher performance, showing the transferring ability on photorealistic datasets. Meanwhile, ProcTHOR fine-tuned on HM3D achieves the best performance. This indicates that there is still a gap between photorealistic and simulated houses.
>
> | Model | SPL↑ | SR↑ |
> |-------|------|-----|
> | ProcTHOR | 31.8 | 54.4 |
> | ProcTHOR (zero-shot) | 7.7 | 13.2 |
> | ZSON | 12.6 | 25.5 |
> | ESC | 22.3 | 39.2 |
> |Our|34.11|41.51|
>
>
>  > Weakness #2: The manual creation of CoT explanation traces for training NATVLM may limit scalability and automation in more complex or dynamic environments.
>
> Thanks for your suggestion. Despite the manual CoT, our NatVLM still performs well thanks to the strong ability of VLMs. We conduct extensive experiments other than DivScene, including iTHOR, ProcTHOR, and HM3D. The results demonstrate that NatVLM can adjust to various environments with high performance.

---

> ### Author Response · Authors · 2024-11-21
> **Response to Reviewer 441m (2/3)**
>
> > Weakness #3: While DIVSCENE offers diverse scenes and objects, the paper lacks a detailed analysis of the scene complexity and its impact on navigation performance. For example, understanding how NATVLM performs in cluttered versus minimal scenes, or in spaces with varying levels of visual occlusion, would add depth to the evaluation. Such insights could help clarify the scenarios where NATVLM’s CoT-based reasoning is most beneficial and identify cases where additional model improvements may be needed.
>
> We provide a detailed comparison of scene complexity between our dataset and the other two datasets: iTHOR and ProcTHOR. From the table below, we can see that our dataset contains many more types of objects, and there are more objects per house.
>
> | Dataset | # total object types | # scene type | # object per house | # object types per house | # rooms per house | Average room size (m^2)|
> |-------|------|-----|-----|-----|-----|-----|
> |ProcTHOR | 38 | 4 | 35.64 | 16.25 | 3.78 | 25.21 |
> |iTHOR | 116 | 4 | 47.26 | 30.92 | 1 | 34.24 |
> |DivScene | 22696 | 81 | 111.42 | 32.17 | 2.35 | 30.15 |
>
> For the impact on navigation performance, we find that cluttered houses with diverse objects and scene types can pose more challenges to navigational agents. Here, we compare the performance of NatVLM on ProcTHOR and our DivScene. We copy the success rate of NatVLM and NatVLM without CoT in the table below from Tables 2 and 5 in our paper. NatVLM (without CoT) shows a significant drop of 4.71 when we move the model from ProcTHOR to DivScene (31.25 -> 26.54). However, with the CoT, the performance of NatVLM only shows a small fluctuation. This observation verifies that diverse scene and object types can pose more challenges to navigation models. Also, our design of CoT-based instruction tuning can help models better tackle those challenges.
>
> | Models | ProcTHOR | DivScene |
> |-------|------|-----|
> | NatVLM | 53.12 | 54.94 |
> | NatVLM (without CoT) | 31.25 | 26.54 |
>
> To be more concrete, we also show two houses with the same number of rooms from DivScene and ProcTHOR, respectively. The images are shown in Appendix E.1 and Figure 9 in our revised paper. Obviously, our scene is more complex with more objects. Quantitively, our scene contains 466, and the scene from ProcTHOR contains only 74 objects.
>
> > Weakness #4: Although the paper emphasizes the effectiveness of NATVLM, it does not address the computational cost of training and deploying the model, especially given the integration of large-scale LVLMs and manually designed CoT traces. Providing a comparison of training and inference times with baseline models would offer readers a clearer understanding of the model’s practicality.
>
> We understand your concern about computational cost. For training, we use 8 x A100 GPUs to train our model. After adding the CoT traces, the training time increases from 10 hours to 17 hours, which is acceptable. Meanwhile, we use 1 A100 GPU to deploy the model. The generation speed increases from 0.28s to 1.03s per query when we add the CoT traces. In practice, we can use more GPUs to speed up the inference. For example, RT-2 [12] uses a multi-TPU cloud service to achieve a frequency of 5 Hz for a 5B model and 3 Hz for a 55B model. We believe the computational cost, both in training and inference, is reasonable in our experiments.

---

> ### Author Response · Authors · 2024-11-21
> **Response to Reviewer 441m (3/3)**
>
> # Reference
> 1. Brohan, Anthony, et al. "Rt-1: Robotics transformer for real-world control at scale." arXiv preprint arXiv:2212.06817 (2022).
> 2. Yang, Yue, et al. "Holodeck: Language guided generation of 3d embodied ai environments." Proceedings of the IEEE/CVF Conference on Computer Vision and Pattern Recognition. 2024.
> 3. Ehsani, Kiana, et al. "SPOC: Imitating Shortest Paths in Simulation Enables Effective Navigation and Manipulation in the Real World." Proceedings of the IEEE/CVF Conference on Computer Vision and Pattern Recognition. 2024.
> 4.  Deitke, Matt, et al. "Objaverse: A universe of annotated 3d objects." Proceedings of the IEEE/CVF Conference on Computer Vision and Pattern Recognition. 2023.
> 5. Savva, Manolis, et al. "Habitat: A platform for embodied ai research." Proceedings of the IEEE/CVF international conference on computer vision. 2019.
> 6. Chang, Angel, et al. "Matterport3d: Learning from rgb-d data in indoor environments." arXiv preprint arXiv:1709.06158 (2017).
> 7. Ramakrishnan, Santhosh K., et al. "Habitat-matterport 3d dataset (hm3d): 1000 large-scale 3d environments for embodied ai." arXiv preprint arXiv:2109.08238 (2021).
> 8. Xia, Fei, et al. "Gibson env: Real-world perception for embodied agents." Proceedings of the IEEE conference on computer vision and pattern recognition. 2018.
> 9. Deitke, Matt, et al. "ProcTHOR: Large-Scale Embodied AI Using Procedural Generation." Advances in Neural Information Processing Systems 35 (2022): 5982-599
> 10. Majumdar, Arjun, et al. "Zson: Zero-shot object-goal navigation using multimodal goal embeddings." Advances in Neural Information Processing Systems 35 (2022): 32340-32352.
> 11. Zhou, Kaiwen, et al. "Esc: Exploration with soft commonsense constraints for zero-shot object navigation." International Conference on Machine Learning. PMLR, 2023.
> 12. Brohan, Anthony, et al. "Rt-2: Vision-language-action models transfer web knowledge to robotic control." arXiv preprint arXiv:2307.15818 (2023).

---

### Author Response · Authors · 2024-12-04
**General Response to All Reviewers**

We sincerely thank all reviewers for their detailed and constructive feedback on our manuscript. **We are delighted that they acknowledge our pioneering contributions and recognize the efforts and strengths of our work:**
1. Open-vocab object navigation is important (Reviewer 441m, YJpB);
2. The method is rigorous, with extensive experiments and ablation study (Reviewer 441m, YJpB);
3. The paper is well-organized with logical structure and clear motivation (Reviewer 441m, YJpB, AM4v);
4. The work is very novel (Reviewer YJpB);
5. Insightful result of CoT traces (Reviewer HDKj);
6. Open sourcing (Reviewer HDKj);
7. Important benchmark with good quality (Reviewer HDKj, AM4v).

**During the rebuttal period, we provide detailed and rigorous responses to every posted comment. The Reviewer YJpB and AM4v raised the score from 5 to 6 and from 3 to 6, respectively, based on our supportive explanations. In those responses, we conduct a series of new experiments to solve the reviewers' concerns:**
1. A thorough comparison with numerous baselines on the photorealistic dataset HM3D.
2. A statistical comparison of our dataset DivScene with previous ones.
3. An analysis of the impact of scene complexity on navigation performance.
4. An analysis of the impact of varying numbers of scene and target object types on the navigation performance.

For the presentation of our work, we want to highlight that our primary objective is to benchmark the current progress of end-to-end LVLMs on the open-vocab object navigation task. Both the dataset and method sections are deliberately designed to serve this central goal, ensuring a cohesive and focused contribution.

Once again, we sincerely thank the reviewers for their thorough evaluation of our manuscript. Their valuable insights have significantly contributed to improving our work. Our comprehensive revisions and responses have successfully addressed the raised concerns and validated the significance of our research contribution.

Best Regards,

Authors from Submission 12096.

---

### Meta-Review · Area_Chair_ki7s · 2024-12-24

**Metareview:**

This paper presents a new benchmark for object navigation, with the goal of increasing diversity of scenes (4.6k) and types (81) leveraging prior GPT-based environmental description generation. The authors then benchmark various LLMs/VLMs on this task, which are trained to imitate the shortest path simulation-based ground truth. The reviewers appreciated the effort in generating the dataset, the development of a LLM/VLM-based approach, and the experimental results. However, the paper is a somewhat confused one, containing a new benchmark (without a clear comparison to prior benchmarks and why it is needed, HDKj) as well as a method (which is not compared to existing approaches to this problem, even if prior methods are closed-set, HDKj/YJpB but also open-vocabulary ones, AM4v). This results in a situation where it is difficult to understand where current methods fail and why the problem is hard (YJpB). Additional concerns included the lack of results in more real-world photorealistic settings, lack of details about the environment generation, etc. The authors provided a rebuttal addressing some of these aspects, comparing the benchmark statistics in a table (though only to THOR-based ones, which is a large limitation as HSSD, etc. are not compared to), comparing the method to other zero-shot methods on HM3D, and scale of performance with respect to object/scene diversity. Some of the reviewers mentioned most of their concerns were addressed.

  Overall, this paper is quite borderline; while three reviewers have positive scores, some still acknowledge the severe limitations of the paper. Some of this stems from the fact that the initial version was severely lacking given that it just proposed a new benchmark without a strong comparison/situation/justification compared to prior ones, and then proposed a method that was not compared to existing state-of-art. While the rebuttal did situate with respect to THOR-based benchmarks, and also did compare to state-of-art methods on HM3D, leading to raised scores, it is still a splintered paper that does not justify why this benchmark is better/harder and how existing methods fail at it (including through analysis), before presenting an approach that addresses those limitations. As a result, I do not recommend acceptance as significant effort is needed to making the paper both coherent and rigorous in terms of the contribution.

**Additional Comments On Reviewer Discussion:**

The reviewers had a number of concerns, especially the weakness of the benchmark with respect to situation/justification and tie to the method, and the weakness of the method in terms of lack of comparisons to existing state-of-art. Some of these were addressed, making the paper stronger, but weaknesses still remained and must be addressed more rigorously for a future submission.

---

### Decision · Program_Chairs · 2025-01-22

Reject